# Long-lived unipotent Blimp1-positive luminal stem cells drive mammary gland organogenesis throughout adult life

Salah Elias [1,2], Marc A. Morgan[1,3], Elizabeth K. Bikoff[1] & Elizabeth J. Robertson[1]

The hierarchical relationships between various stem and progenitor cell subpopulations driving mammary gland morphogenesis and homoeostasis are poorly understood. Conditional inactivation experiments previously demonstrated that expression of the zinc finger transcriptional repressor Blimp1/PRDM1 is essential for the establishment of epithelial cell polarity and functional maturation of alveolar cells. Here we exploit a *Prdm1.CreERT2-LacZ* reporter allele for lineage tracing experiments. Blimp1 expression marks a rare subpopulation of unipotent luminal stem cells that initially appear in the embryonic mammary gland at around E17.5 coincident with the segregation of the luminal and basal compartments. Fate mapping at multiple time points in combination with whole-mount confocal imaging revealed these long-lived unipotent luminal stem cells survive consecutive involutions and retain their identity throughout adult life. Blimp1$^+$ luminal stem cells give rise to Blimp1$^-$ progeny that are invariably Elf5$^+$ER$\alpha^-$PR$^-$. Thus, Blimp1 expression defines a mammary stem cell subpopulation with unique functional characteristics.

[1] Sir William Dunn School of Pathology, University of Oxford, Oxford OX1 3RE, UK. [2] Biological Sciences, University of Southampton, Life Sciences Building 85, University Road, Southampton SO17 1BJ, UK. [3] Present address: Department of Biochemistry and Molecular Genetics, Northwestern University Feinberg School of Medicine, Chicago, IL 60611, USA. Correspondence and requests for materials should be addressed to S.E. (email: S.K.Elias@soton.ac.uk) or to E.J.R. (email: elizabeth.robertson@path.ox.ac.uk)

Postnatal morphogenesis of the mammary gland in response to hormonal stimuli, sets the stage for the dramatic tissue turnover and remodelling seen during successive rounds of pregnancy[1]. The mammary epithelium is composed of two distinct cell populations: the outer myoepithelial/basal cells and the inner luminal cells[1]. During pregnancy, this network of highly branched ducts massively expands giving rise to the specialised milk-secreting alveoli. Subsequently as the newborn pups undergo the suckling-weaning transition the glands regress, a process termed involution. Repeated rounds of tissue morphogenesis during successive pregnancies reflect the regenerative capabilities of mammary stem cells. Reconstitution studies have shown that an entire functional mammary gland can be generated from the progeny of a single basal cell, thought to represent a common bipotent stem cell[2, 3]. On the other hand, in vivo lineage tracing studies challenge the existence of bipotent stem cells during postnatal development and argue that stem cells are restricted to either the luminal or myoepithelial compartment[4–6]. A likely possibility is that multiple highly dynamic stem/progenitor cells collectively contribute to the mammary epithelial hierarchy. Several unipotent basal and luminal progenitor cell subsets have been characterised[5–12]. Rare bipotent basal stem cell subsets with dynamic developmental potential have also been identified[8, 13]. Thus the signalling pathways and transcriptional regulators that instruct postnatal progenitors to become lineage-restricted remain ill defined.

Within the luminal compartment, several distinct cell subsets have been described to display distinct differentiation states and developmental potential[9, 10, 14–16]. Functionally mature Oestrogen receptor-positive (ERα[+]) luminal cells display low proliferative capacity[12, 15, 17]. By contrast, ERα[−] luminal cells that robustly express the Ets transcription factor Elf5 are highly proliferative progenitors[5, 9–11, 16, 18]. Rare subsets of highly proliferative luminal progenitors, heterogeneous for progesterone receptor (PR) and ERα expression, have also been identified[9, 12]. During pregnancy, hormone responsive ERα[+] and PR[+] luminal cells induce the proliferation of neighbouring ERα[−] and PR[−] cells to drive alveologenesis[19]. Recent evidence strongly suggests that these luminal sub-sets may represent the cell types of origin for heterogeneous and aggressive breast tumours[20–22]. Unravelling the hierarchical relationships between these luminal stem cell populations remains an important priority.

The PR/SET domain zinc finger transcriptional repressor Blimp1, a member of the Prdm family, governs numerous cell fate decisions in the developing embryo and adult tissues[23]. Previous studies have described critical roles during primordial germ-cell specification[24, 25], placental morphogenesis[26, 27], regulation of postnatal intestinal maturation[28, 29], and maintenance of tissue homoeostasis and epithelial barrier function in adult skin[30, 31]. We recently identified a rare subset of Blimp1-expressing luminal cells in the postnatal mammary gland. Blimp1 is robustly induced in the alveoli during pregnancy, and conditional inactivation experiments revealed Blimp1 function is essential for functional maturation of the forming alveoli[32]. Here we exploit a Prdm1Cre[ERT2] reporter mouse strain to examine the possible relationships between Blimp1-expressing cells and previously described luminal progenitor cell sub-populations. Lineage tracing experiments were used to evaluate their potentially dynamic contributions during mammary gland morphogenesis and tissue homoeostasis. We demonstrate that Blimp1[+] cells, initially detectable at embryonic (E) E17.5 in mammary rudiments, represent lineage-restricted, unipotent luminal progenitors that invariably lack ERα and PR expression. While Blimp1[+] cells represent a very rare subset of luminal progenitors they display high self-renewal capacity, and contribute extensively to duct formation and homoeostasis, and to alveologenesis during

pregnancy. Moreover, long-lived Blimp1[+] luminal progenitors, specified during embryogenesis, survive multiple rounds of pregnancy and involution. Collectively the present experiments demonstrate that Blimp1 expression marks a unipotent luminal stem cell population that substantially contributes to mammary gland morphogenesis throughout adult life.

## Results

**Prdm1/Blimp1Cre[ERT2] allows lineage tracing of Blimp1[+] cells.**
We previously exploited a Blimp1-mVenus BAC transgenic reporter strain[33], to identify a subset of highly clonogenic luminal cells[32]. To further characterise the contributions of Blimp1[+] luminal cells during mammary gland development and homoeostasis, here we engineered a Prdm1.Cre[ERT2].LacZ reporter allele (Prdm1Cre[ERT2].LacZ) containing a Cre[ERT2]-IRES-nLacZ cassette in the first coding exon of the Prdm1 locus (Supplementary Fig. 1a, b). To confirm that expression of the nuclear LacZ reporter cassette faithfully recapitulates dynamic patterns of Blimp1 expression in the early embryo[25, 27] we crossed Prdm1Cre[ERT2].LacZ/+ males to wild-type females and stained the resulting embryos. As expected, at E7.5, LacZ activity is confined to Blimp1[+] PGCs emerging at the proximal end of the primitive streak (Supplementary Fig. 1c). Later, at E9.5, E10.5 and E12.5 (Supplementary Fig. 1d–f), LacZ staining precisely marks previously described sites of Blimp1 expression within the second heart field, limb buds and otic vesicle[25, 27].

Next, Prdm1Cre[ERT2/+] mice were crossed to the Rosa26[mTmG] (R26R[mTmG]) reporter strain[34, 35] to generate Prdm1Cre[ERT2/+]; R26R[mTmG/+] females. In this context, administration of tamoxifen (TAM) leads to nuclear accumulation of Cre recombinase and concomitant excision of the floxed reporter locus. Consequently expression of membrane-bound tdTomato is switched to membrane-bound green fluorescent protein (GFP) (Supplementary Fig. 1g). The ability of Prdm1Cre[ERT2] to mark progenitor cells was confirmed by administering a very low single dose of TAM (0.5 mg/25 g body weight) to pregnant Prdm1Cre[ERT2/+]; R26R[mTmG/+] females at pregnancy day (P) P15.5, and examining expression in the embryonic intestinal epithelium 24 h later. As expected at E16.5 the progeny of highly proliferative GFP[+]Blimp1[+] intervillous pocket cells were detectable along the forming villus axis (Supplementary Fig. 1h)[28]. Similarly GFP expression in postnatal skin identifies the sebaceous gland and dermal papillae of the hair follicles as well as a subset of granular layer keratinocytes (Supplementary Fig. 1i)[27, 31]. Thus, the Prdm1Cre[ERT2] reporter allele faithfully marks Blimp1[+] cell lineages.

**Unipotent Blimp1[+] stem cells initiate in the mammary rudiment.** To detect the onset of Blimp1 expression in the embryonic mammary gland, we administered a single dose of TAM (0.5 mg/25 g body weight) to pregnant Prdm1Cre[ERT2/+];R26R[mTmG/+] females at P12.5, P14.5 or P17.5, and examined the embryonic mammary placodes after 24 h (Fig. 1a). At E13.5 and E15.5 GFP[+] cells were restricted to the surrounding mesenchyme (Fig. 1b). By contrast at E18.5 a few GFP[+] cells detectable within the mammary rudiments exclusively express the luminal (K8) but not basal (K14) marker (1.5 ± 0.2% of the total K8[+] cell population) (Fig. 1b, c). The majority of GFP[+] cells (77.3 ± 3.5%) expressed Blimp1 at this stage (Fig. 1d). Thus the onset of Blimp1 expression coincides with the basal vs. luminal cell fate decision that occurs at around E17.5[36].

To assess contributions made by the embryonic Blimp1[+] cell lineage during postnatal mammary gland development, we injected P17.5 pregnant females with a single dose of TAM (Fig. 1e), and examined Prdm1Cre[ERT2/+];R26R[mTmG/+] female

offspring once they reached adulthood (11 weeks). The SeeDB (See Deep Brain) tissue-clearing protocol[37, 38], was optimised to allow three-dimensional (3D) confocal imaging in the intact, undisturbed mammary gland. Remarkably, we were able to identify large GFP⁺ cell clones that retain their luminal identity

throughout the mammary ductal network, including most distal tips (Fig. 1f, g). Complementary flow cytometry analysis, using CD24 and CD49f surface markers to resolve the luminal (CD24$^{high}$CD49f$^{low}$) and myoepithelial (CD24$^{low}$CD49f$^{high}$) cell compartments, confirmed that GFP⁺ progeny were exclusively

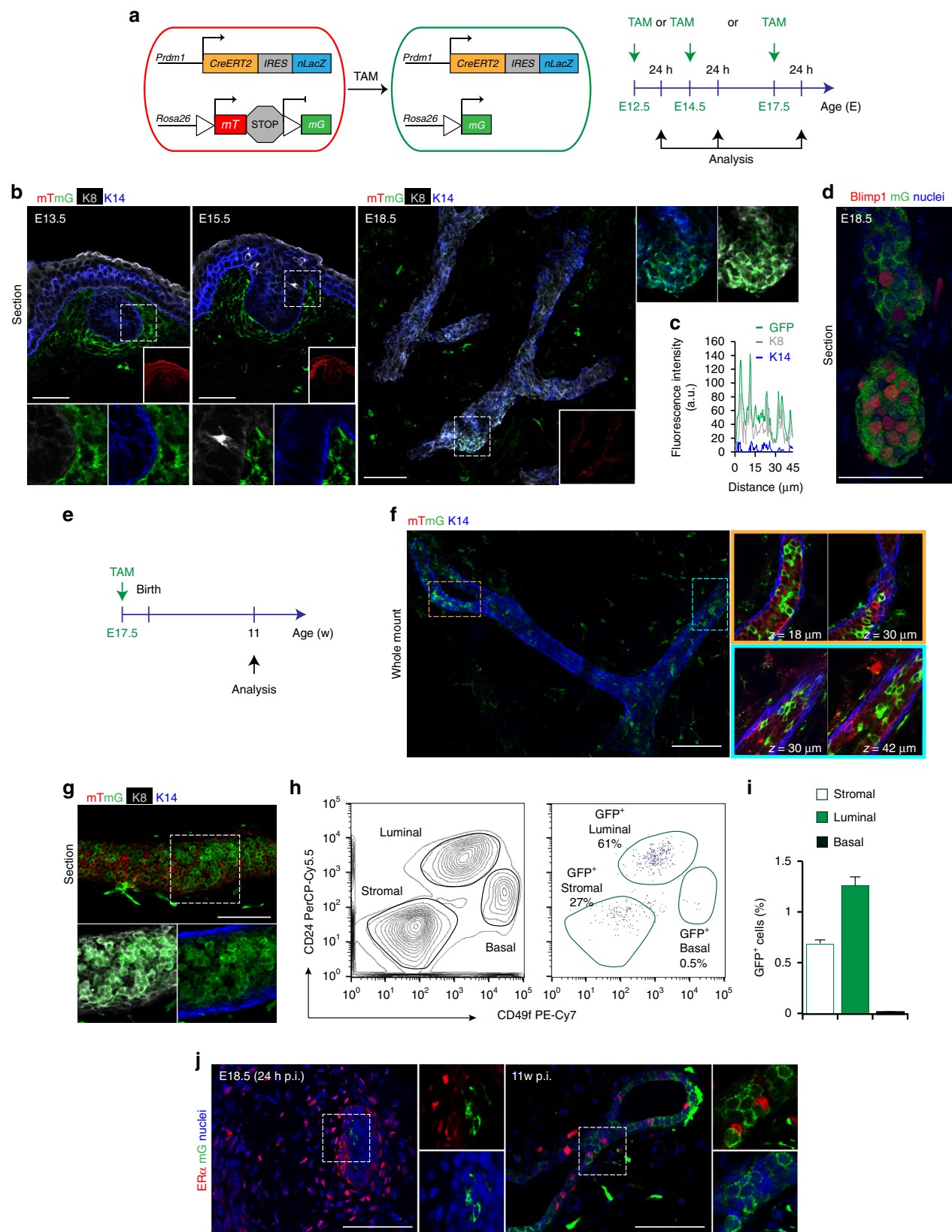

restricted to the luminal subset, representing $1.26 \pm 0.1\%$ of the total luminal cell population (Fig. 1h, i). Interestingly GFP$^+$ cells invariably lacked ERα expression throughout embryonic and postnatal mammary gland development (Fig. 1j) allowing us to conclude that Blimp1 expression in the embryonic mammary rudiment marks a subset of unipotent ERα$^-$ luminal stem cells.

**Blimp1$^+$ stem cells drive ductal morphogenesis and homoeostasis.** To examine the fate and positions of the progeny of single labelled Blimp1$^+$ cells, here we treated pubertal and adult virgin $Prdm1Cre^{ERT2/+};R26R^{mTmG/+}$ females with a single low dose of TAM (4 mg/25 g body weight; Fig. 2, Supplementary Fig. 2). Additionally to visualise deep regions at high resolution in whole-mount cleared glands, we used a confocal microscope equipped with a 40x silicon oil-immersion Super Apochromatic lens (UPLSAPO 40XS). Compared to conventional oil immersion objectives, this objective significantly reduces loss of contrast due to spherical aberration and provides higher resolution and brightness, especially when imaging thick samples. Together these technical advances allowed us to easily visualise single GFP cell clones at 24h-post TAM induction. Twenty-four hours after TAM administration, we observed rare GFP$^+$ single luminal cells ($0.25 \pm 0.05\%$, puberty, Fig. 2a, b, e; and $0.2 \pm 0.03\%$, adult virgin; Supplementary Fig. 2a, b, e). These frequencies are in complete agreement with those we reported previously using the Blimp1-mVenus BAC reporter line[32], indicating the protocol very efficiently labels Blimp1$^+$ single-cells.

Next, we performed pulse-chase experiments to evaluate clonal expansion of Blimp1$^+$ single cells during ductal morphogenesis. Pubertal females were treated with TAM at 4 weeks of age and analysed at different time points up to 24 weeks of age (Fig. 2c, d). We found that $75 \pm 4.6\%$ of GFP$^+$ single-cell clones expand giving rise to multicellular clones, which markedly increase in size over time. Only $25 \pm 3.2\%$ of GFP$^+$ single cells remain quiescent over the course of pubertal development. Previous quantification of the clonal expansion of K8$^+$ luminal cells during this same developmental window, indicates that only 40% of the labelled cells give rise to clones[4]. The $Prdm1Cre^{ERT2/+}$ reporter allele thus identifies a discrete sub-set of highly clonogenic luminal progenitor cells. Consistent with this, the frequencies of GFP$^+$ cells within the total luminal population increased significantly with time (Fig. 2e). Close examination of the labelling patterns revealed large multicellular clones present in primary and secondary ducts (Fig. 2c). These regions contained several hundred GFP$^+$ cells spanning numerous contiguous branching ducts (Fig. 2c). Additionally, we detected isolated regions containing either limited numbers of GFP$^+$ or GFP$^-$ cells (Fig. 2c),

further validating our efficient single-cell labelling 24 h post TAM. Identical labelling patterns were observed in adult virgin $Prdm1Cre^{ERT2/+};R26R^{mTmG/+}$ females induced at 9 weeks of age (Supplementary Fig. 2). GFP$^+$ luminal cells displayed high clonal and proliferative capacities up to 29 weeks of age, and actively contribute to ductal homoeostasis (Supplementary Fig. 2). Collectively, these experiments show that Blimp1$^+$ luminal cells give rise to progeny that contribute substantially to both ductal morphogenesis and homoeostasis.

In both pubertal and adult virgin $Prdm1Cre^{ERT2/+};R26R^{mTmG/+}$ females, the vast majority of luminal Blimp1$^+$ cells were GFP$^+$ 24 h post TAM induction (pubertal: $99 \pm 0.5\%$, Fig. 2f, g; adult virgin: $99 \pm 0.8\%$, Supplementary Fig. 2f, g). We observed only a very small number (~1% of total Blimp1$^+$ cells) classified as Blimp1$^+$GFP$^-$ cells which expressed cytoplasmic GFP, and notably none were totally devoid of GFP staining. Consistent with this, all Blimp1$^+$ progeny expressed membrane-bound GFP 20 weeks after TAM induction. Interestingly, in both pubertal and adult virgin females, double staining experiments showed that Blimp1$^+$ luminal cells give rise to GFP$^+$ progeny that mostly lack Blimp1 expression 20 weeks after TAM induction (pubertal: $93.4 \pm 4.0\%$, Fig. 2f, h; adult virgin: $86.2 \pm 5.5\%$, Supplementary Fig. 2f, h). Without exception the small numbers of Blimp1$^+$ cells observed are GFP$^+$, and we failed to detect Blimp1$^+$ cells lacking GFP expression at any stage examined. Thus Blimp1 expression identifies a rare but persistent population of luminal progenitors that give rise to highly self-renewing Blimp1$^-$ progeny.

**Blimp1$^+$ stem cells give rise to Elf5$^+$ERα$^-$PR$^-$ luminal progenitors.** Our recent experiments demonstrate that Blimp1$^+$ luminal cells express Elf5 but lack ERα and PR[32]. Interestingly conditional loss-of-function of Blimp1 in $K14$-$Cre;Prdm1^{CA/BEH}$ virgin females markedly reduces the number of Elf5$^+$ cells (Supplementary Fig. 3). To investigate possible contributions of Blimp1$^+$ progeny to the Elf5$^+$ERα$^-$PR$^-$ lineage, we next examined adult virgin $Prdm1Cre^{ERT2/+};R26R^{mTmG/+}$ female mice 24 h after TAM administration (Fig. 3a). As assessed by flow cytometry analysis GFP$^+$ cells represent $0.22 \pm 0.03\%$ of the total luminal population (CD24$^{high}$CD49f$^{low}$) (Fig. 3b, c). As expected the GFP$^+$ Blimp1$^+$ cells express Elf5, but lack ERα and PR expression (Fig. 3d). Eight weeks after TAM administration, GFP$^+$ luminal clones continue to express Elf5 but remain ERα$^-$PR$^-$ (Fig. 3d). Interestingly, we observed numerous GFP$^+$ cells surrounded by clusters of GFP$^-$ERα$^+$PR$^+$ cells (Fig. 3d). At this stage, clonal expansion of the GFP$^+$ERα$^-$PR$^-$ cells has occurred at the expense of the GFP$^-$ERα$^-$PR$^-$ cell population suggesting that the Blimp1$^+$ stem cells contribute extensively to the ERα$^-$PR$^-$ lineage.

---

**Fig. 1** Blimp1 marks unipotent embryonic luminal stem cells. **a** Lineage tracing strategy adopted in **b-d**. $Prdm1Cre^{ERT2}$ mice are crossed to $R26R^{mTmG}$ reporter mice. Pregnant females were injected intraperitoneally with tamoxifen (TAM, 0.5 mg/25 g body weight) at embryonic day (E) E12.5, E14.5 or E17.5. Double transgenic littermates $Prdm1Cre^{ERT2/+};R26R^{mTmG/+}$ were analysed 24 h later. **b** Cryosections (100 μm thick) from E13.5, E15.5 and E18.5 embryonic mammary glands stained for GFP (green), keratin 8 (K8, grey) and keratin 14 (K14, blue). mT is displayed in red ($n = 4$ embryos). Scale bars, 50 μm. **c** Representative line scan-analysis (representative fluorescence intensity, minimum 20 cell clusters analysed). **d** Cryosections (100 μm thick) from E18.5 embryonic mammary glands stained for GFP (green) and Blimp1 (red) and counterstained with DAPI (Nuclei, blue). **e** Lineage tracing strategy adopted in **f–j**. Females are injected with TAM at pregnancy day (P) P17.5. Double transgenic littermates $Prdm1Cre^{ERT2/+};R26R^{mTmG/+}$ were analysed 11 weeks (11w) after birth. **f** 3D imaging of SeeDB-cleared 11-week-old adult virgin mammary tissue. Glands are stained for GFP (green) and K14 (blue). mT is displayed in red ($n = 3$ mice). Scale bars, 100 μm. **g** Cryosections (50 μm thick) from 11-week-old adult virgin mammary glands stained for GFP (green), K8 (grey) and K14 (blue) ($n = 3$ mice). Scale bar, 50 μm. **h** FACS analysis of MECs from 11-week-old adult virgin mammary glands. CD45$^+$ leukocytes and CD31$^+$ endothelial cells were removed and total epithelial cells gated on a CD24/CD49f plot. Left: FACS plot showing discrimination of basal (CD24$^{low}$CD49f$^{high}$) and luminal (CD24$^{high}$CD49f$^{low}$) cell populations. Right: FACS plot showing distribution of GFP$^+$ cells within stromal, luminal and basal cell populations. ($n = 6$ mice pooled from two independent experiments). **i** Percentages of GFP$^+$ stromal (0.65% GFP$^+$ cells), luminal (1.26% GFP$^+$ cells) and basal (0.02% GFP$^+$ cells) cell populations. Data are presented as mean ± s.e.m. ($n = 6$ mice pooled from two independent experiments). **j** Cryosections (20 μm thick) from E18.5 rudiments and 11-week-old adult virgin mammary glands stained for GFP (green), ERα (red) and counterstained with DAPI (Nuclei, blue). ($n = 3$ mice). Scale bars, 50 μm

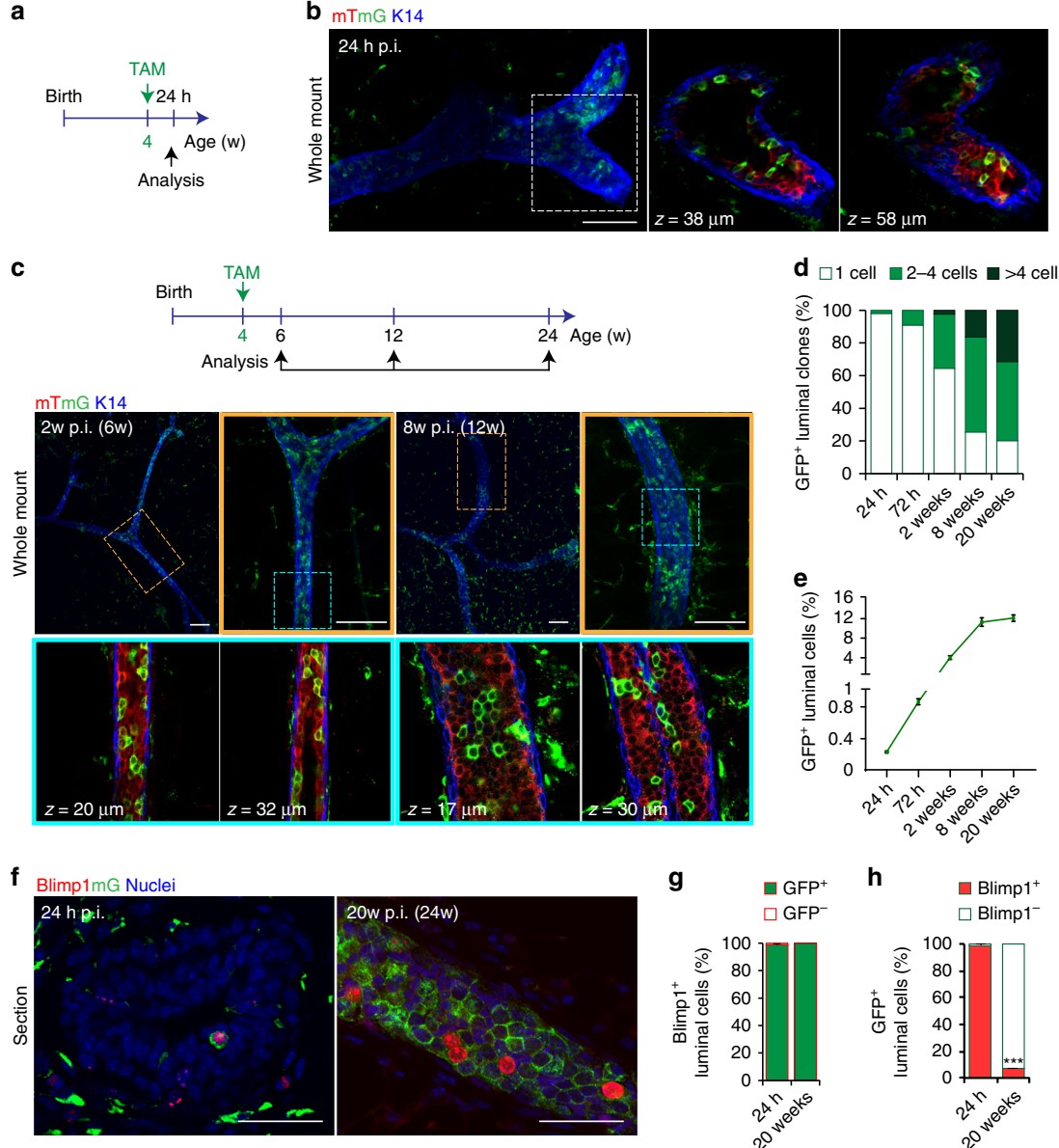

**Fig. 2** Blimp1-derived cells represent luminal progenitors that contribute actively to ductal morphogenesis. **a** Lineage tracing strategy adopted in **b**, **d–h**. Four-week-old pubertal *Prdm1Cre^(ERT2/+)*;*R26R^(mTmG/+)* females were injected with TAM (4 mg/25 g body weight), and analysed 24 h later. **b** 3D imaging of SeeDB-cleared 4-week-old pubertal mammary tissue. Glands are stained for GFP (green) and K14 (blue). mT is displayed in red (*n* = 4 mice). Scale bar, 100 μm. **c** Lineage tracing strategy adopted in **c–h**. Four-week-old pubertal *Prdm1Cre^(ERT2/+)*;*R26R^(mTmG/+)* females injected with TAM (4 mg/25 g body weight), were analysed at 2w, 8w or 20w post induction (p.i.), at 6w, 12w or 24w after birth respectively. 3D imaging of SeeDB-cleared of mammary tissue at 2w and 8w p.i. in puberty. Glands are stained for GFP (green) and K14 (blue). mT is displayed in red. (*n* = 4: 2w p.i.; *n* = 3: 8w p.i.). Scale bars, 100 μm. **d** Percentages of GFP+ clones containing 1, 2–4 and >4 cells at different time points post-TAM induction. Data are presented as mean ± s.e.m. (number of clones pooled from *n* = 3 mice per time point: *n* = 400, *n* = 374, *n* = 453, *n* = 393, *n* = 508, respectively). ***P < 0.001 (one-way ANOVA). **e** Percentages in puberty of GFP+ cell within the total luminal cell population, at different time points post-TAM induction. Data are presented as mean ± s.e.m. (number of GFP+ cells pooled from *n* = 3 mice per time point: *n* = 791, *n* = 3172, *n* = 4149, *n* = 7390, *n* = 7956, respectively). ***P < 0.001 (one-way ANOVA). **f** Cryosections (20 μm thick) from mammary glands at 24 h and 20w post-TAM induction (p.i.) in puberty, stained for GFP (green), Blimp1 (red) and counterstained with DAPI (Nuclei, blue). (*n* = 3 mice per time point). Scale bars, 50 μm. **g** Percentages in puberty of Blimp1+GFP+ vs. Blimp1+GFP−cells at 24 h and 20w post-TAM induction. Data are presented as mean ± s.e.m. (Number of Blimp1+ cells pooled from *n* = 3 mice per time point: *n* = 262 and *n* = 290, respectively). P > 0.05 (t-test). **h** Percentages in puberty of GFP+Blimp1+ vs. GFP+Blimp1− cells at 24 h and 20w post-TAM induction. Data are presented as mean ± s.e.m. (number of GFP+ cells pooled from *n* = 3 mice per time point: *n* = 275 and *n* = 2214, respectively). ***P < 0.001 (t-test)

Collectively these results strongly suggest that the progeny of Blimp1+ luminal progenitors are lineage-restricted and retain their Elf5+ERα−PR− identity throughout development.

Next, to examine the proliferative status of GFP+ luminal cells we assessed Ki67 expression. Twenty-four hours after TAM induction, 33 ± 1.5% of Ki67+ cells were GFP+, and this population expanded to reach 58.2 ± 2.7% 8 weeks after tamoxifen induction (Fig. 3e, f). To directly examine their proliferative capabilities, sorted GFP+ luminal cells collected from 24 h TAM-treated adult virgin *Prdm1Cre^(ERT2/+)*;*R26R^(mTmG/+)* females, were seeded in 3D Matrigel cultures (Fig. 3g, h). The colony-forming activity of the GFP+ cell fraction was markedly

enriched; $35 \pm 2.5\%$ of plated GFP$^+$ luminal cells formed colonies compared with $17 \pm 2.3\%$ of the starting luminal cell population, where GFP$^+$ colonies represented ~59% of the total. In contrast only ~7% of GFP$^-$ luminal cells exhibited colony-forming potential (Fig. 3g, h). Thus Blimp1$^+$ stem cells give rise to highly clonogenic Elf5$^+$ER$\alpha^-$PR$^-$ luminal progenitors.

**Blimp1$^+$ cells represent long-lived alveolar progenitors**. Elf5 represents an alveolar lineage-specific master regulator, and Elf5 mutant mammary glands completely fail to initiate alveologenesis[39]. The ER$\alpha^-$ luminal subpopulation, characterised by robust Elf5 expression[5, 10, 18], is highly responsive to progesterone, via paracrine Rankl signalling from the adjacent ER$\alpha^+$PR$^+$ cell

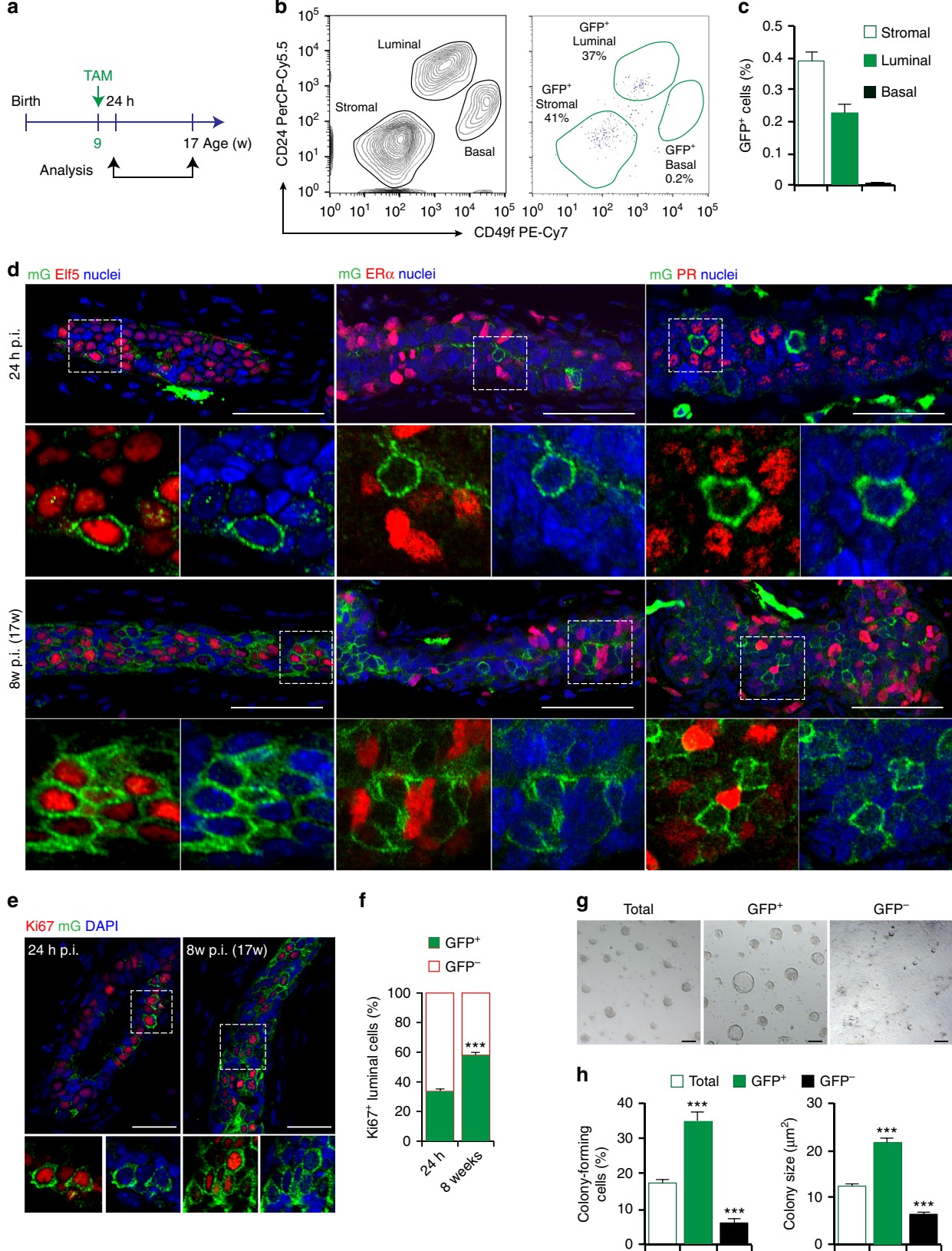

population[40]. This interaction promotes alveolar cell expansion and functional maturation during pregnancy[9, 10, 40]. The results above strongly suggest that Blimp1[+] cells give rise to highly proliferative Elf5[+]ERα[−]PR[−] luminal progenitors that dramatically expand during pregnancy to promote alveologenesis. To directly test this possibility, we pulse-labelled Blimp1[+] cells in pubertal *Prdm1Cre[ERT2/+];R26R[mTmG/+]* females and examined GFP[+] progeny after three consecutive pregnancies and during involution (Fig. 4a). GFP[+]Blimp1 progeny massively expand during pregnancy giving rise to large numbers (22.2 ± 1.8%) of GFP[+] alveoli that for the most part are exclusively comprised of GFP[+] luminal cells (65.6 ± 1.6%) (Fig. 4b–d). Moreover, we found the percentage of GFP[+] alveoli remained constant (21.9 ± 1.5%) after three successive pregnancies (Fig. 4b–d). Upregulated Blimp1 expression during pregnancy is essential for alveolar maturation and efficient milk secretion[32]. Here we observe during pregnancy that the numbers of Blimp1[+]GFP[+] cells remained constant (P1; 13.5 ± 0.6% vs. P3; 14 ± 0.8%) (Fig. 4e, f). By contrast, during involution the majority of surviving residual GFP[+] luminal cells were Blimp1[+] (58 ± 1.8%) (Fig. 4e, f). Importantly, we failed to detect any Blimp1[+] cells lacking GFP which would result from de novo Blimp1 expression induced during involution. Thus we conclude that this Blimp1[+] cell population is derived exclusively from the rare Blimp1[+] cells originally labelled during puberty. Blimp1[+] cells are thus maintained as a constant pool of long-lived alveolar progenitors throughout pregnancy-induced mammary gland remodelling, and survive multiple rounds of involution.

To further characterise responses of Blimp1-expressing cells to pregnancy hormones, we cultured fragments of mammary epithelia from adult *Prdm1Cre[ERT2/+];R26R[mTmG/+]* mice. The 3D organoids were pulsed with TAM for 6 h, and then treated with 17β-estradiol (2E) and progesterone (Pg), or Rankl for 1 or 7 days (Supplementary Fig. 4). Expansion of GFP[+] cells was observed in both cases but Rankl treatment resulted in the most dramatic response (Supplementary Fig. 4a, b). These ex vivo results confirm that Blimp1[+]-derived cells within the luminal compartment respond correctly to pregnancy signalling and are responsive to paracrine Rankl signals from neighbouring ERα[+]PR[+] cells.

## Discussion

The functional relationships and behavioural dynamics of various stem and progenitor cell populations that contribute to mammary gland development and tissue homoeostasis remain poorly understood[1, 19]. Multipotent stem cells are known to be present at embryonic stages[4, 10, 13, 41], including a sub-population that remains bipotent in the postnatal mammary gland[5, 13]. Moreover subsets of multipotent stem cells have been shown to restrict their lineage potential postnatally and contribute to either the basal or luminal compartments[4, 10, 41]. Unipotent stem cells present in the mammary buds capable of switching their potential between multipotency and unipotency at distinct postnatal developmental stages, have also been characterised[8]. The present experiments describe for the first time a very rare subset of Blimp1[+] lineage-restricted luminal stem cells detectable within the embryonic mammary epithelium that maintain their identity throughout adult life, display extensive self-renewal capacity, and contribute to duct morphogenesis, homoeostasis and alveologenesis during consecutive pregnancies (Fig. 5).

Our lineage tracing studies demonstrate that this Blimp1[+] stem cell population is specified during the later stages of embryonic mammary gland development (E17.5–E18.5) coincident with the segregation of the basal and luminal compartments[36] but is exclusively restricted to the luminal compartment. Blimp1[+] cells that initially emerge as small isolated cell clusters within the embryonic rudiment represent a discrete developmentally restricted population within the luminal epithelium that is already committed to the ERα[−] cell lineage. It is well known that Bmp/Smad signals activate Blimp1 expression in the small number of epiblast cells that become exclusively allocated to the PGC lineage in the early post-implantation embryo[42]. It will be interesting to characterise the highly localised upstream signalling pathways responsible for induction of Blimp1 expression in this discrete population of mammary stem cells.

Adult luminal cell populations have been shown to display heterogeneous ERα expression[1]. The Lgr6[+] and Notch3[+] subpopulations that are heterogeneous for ERα expression undergo a temporal state of quiescence[9, 12]. In contrast highly clonogenic Notch1[+] luminal progenitors are invariably ERα[− 10]. These progenitors seem to represent distinct ERα[+] and ERα[−] cell lineages that exclusively give rise to ERα[+] and ERα[−] daughter cells, respectively[9]. In keeping with this, recent studies using an inducible *ERα.Cre* transgenic strain show that the ERα[+] luminal cells derive from ERα[+] stem cells that emerge postnatally. Long-term lineage tracing demonstrate these stem cells are lineage restricted and give rise exclusively to ERα[+] progeny during homoeostasis and regeneration[43]. Nonetheless, despite their differing ER signalling abilities these discrete luminal progenitor sub-sets seem to make equivalent contributions to both duct formation and alveologenesis. Blimp1 conditional inactivation reduces the numbers of Elf5[+] cells and increases the number of ERα[+] cells within the luminal compartment[32]. Interestingly, we found here that Blimp1[+] progenitors mostly give rise to daughter cells

**Fig. 3** Blimp1-derived cells represent highly clonogenic and long-lived Elf5[+]ERα[−]PR[−] luminal progenitors. **a** Lineage tracing strategy adopted in **b–f**. Nine-week-old adult virgin *Prdm1Cre[ERT2/+];R26R[mTmG/+]* females were injected with TAM (4 mg/25 g body weight) and analysed 24 h or 8w later (24 h or 8w p. i.). **b** FACS analysis of 9-week-old adult virgin mammary gland MECs at 24 h post-TAM induction. CD45[+] leukocytes and CD31[+] endothelial cells were removed and total epithelial cells gated on a CD24/CD49f plot. Left: FACS plot showing discrimination of basal (CD24[low]CD49f[high]) and luminal (CD24[high]CD49f[low]) cell populations. Right: FACS plot showing distribution of GFP[+] cells within stromal, luminal and basal cell populations (n = 6 mice pooled from two independent experiments). **c** Percentages of GFP[+] stromal (0.39% GFP[+] cells), luminal (0.22% GFP[+] cells) and basal (0.008% GFP[+] cells) cell populations. Data are presented as mean ± s.e.m. (n = 6 mice pooled from two independent experiments). **d** Cryosections (50 μm thick) from adult virgin mammary glands analysed at 24 h and 8w post-TAM induction (p.i.) stained for GFP (green), Elf5 or ERα or PR (red) and counterstained with DAPI (Nuclei, blue) (n = 3 mice per time point). Scale bars, 50 μm. **e** Cryosections (20 μm thick) from adult virgin mammary glands analysed at 24 h and 8w post-TAM induction (p.i.) stained for GFP (green), Ki67 (red) and counterstained with DAPI (Nuclei, blue) (n = 3 mice per time point). Scale bars, 20 μm. **f** Percentages of Ki67[+]GFP[+] vs. Ki67[+]GFP[−] cells in adult virgin females at 24 h and 8w post-TAM induction. Data are presented as mean ± s.e.m. (number of Ki67[+] cells pooled from n = 3 mice per time point: n = 2575 and n = 3219, respectively). ***P < 0.001 (t test). **g** Photographs of colonies formed by GFP[+], GFP[−] and total luminal cells isolated from 9-week-old adult virgin mice, at 24 h post-TAM induction (p.i.) (n = 6 pooled mice from two independent experiments). **h** Left: colony-forming efficiency of GFP[+], GFP[−] and total luminal cells isolated from 9-week-old adult virgin mice, at 24 h post-TAM induction (p.i.). Right: quantification of colony size of formed by GFP[+], GFP[−] and total luminal cells. Data are presented as mean ± s.e.m. (n = 6 pooled mice from two independent experiments). ***P < 0.001. Scale bars, 50 μm

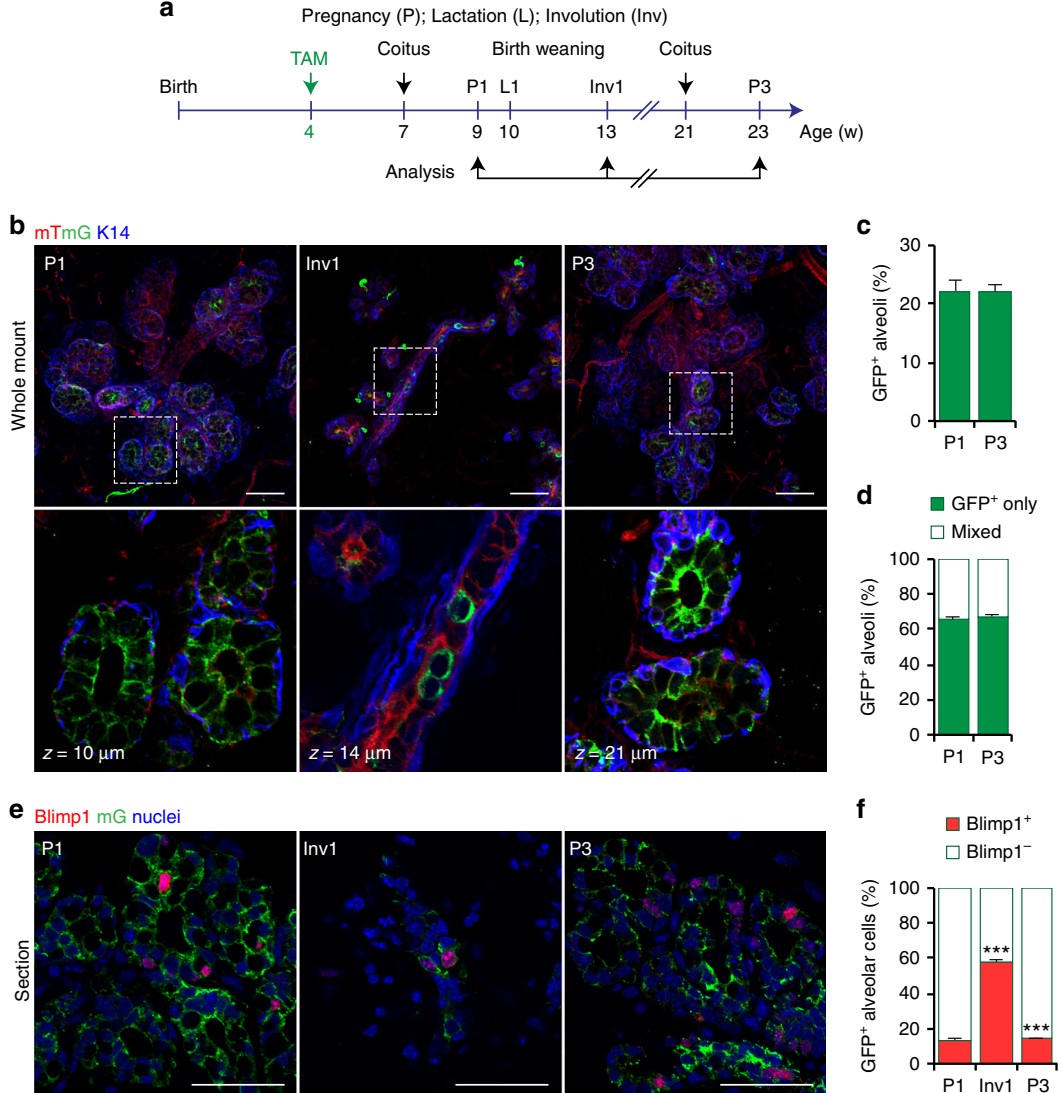

**Fig. 4** Blimp1-derived cells represent long-lived luminal stem cells that contribute actively to consecutive pregnancies. **a** Lineage tracing strategy adopted in **b–f**. Four-week-old adult virgin *Prdm1Cre*[ERT2/+]*;R26R*[mTmG/+] females were injected with TAM (4 mg/25 g body weight) and analysed at day 14.5 of the first and third pregnancy (P1 and P3), and during day 14 of involution (Inv1). **b** 3D imaging of SeeDB-cleared P1, Inv1 and P3 mammary tissue. Glands are stained for GFP (green) and K14 (blue). mT is displayed in red. (n = 3 mice for each time point). Scale bars, 50 μm. **c** Percentages of GFP⁺ alveoli in P1 and P3 mammary glands. Data are presented as mean ± s.e.m. (number of GFP⁺ alveoli pooled from n = 3 mice per time point: n = 534 and n = 561, respectively). P > 0.05 (t-test). **d** Percentages of GFP⁺ alveoli at P1 and P3 containing either exclusively GFP⁺ cells or mixed populations of GFP⁺ and GFP⁻ cells. Data are presented as mean ± s.e.m. (number of GFP⁺ alveoli pooled from n = 3 mice per time point: n = 534 and n = 561, respectively). P > 0.05 (t-test). **e** Cryosections (20 μm thick) from P1, Inv1 and P3 mammary glands analysed stained for GFP (green), Blimp1 (red) and counterstained with DAPI (Nuclei, blue) (n = 3 mice per time point). Scale bars, 20 μm. **f** Percentages of GFP⁺Blimp1⁺ vs. GFP⁺Blimp1⁻ cells at P1, Inv1 and P3. Data are presented as mean ± s. e.m. (number of GFP⁺ cells pooled from n = 3 mice per time point: n = 2068, n = 184 and n = 2256, respectively). ***P < 0.001 (t-test)

lacking Blimp1 expression that are invariably ERα⁻Elf5⁺. However although Blimp1-derived progeny contribute extensively to the ERα⁻ lineage they only give rise to around a fifth of the alveolar structures during pregnancy. This observation, as well as the presence of GFP⁻ERα⁻PR⁻ luminal cells implicates the existence of additional stem cell population(s) that also contribute to the ERα⁻ lineage. Collectively our experiments suggest that Blimp1 expression marks a reserve pool of ERα⁻Elf5⁺ luminal stem cells. Additional work will be required to define these complex lineage relationships and identify further luminal stem cells.

Ductal expansion during puberty has been proposed to depend on the combined activities of lineage restricted stem cells present in the TEBs, that function co-ordinately as equipotent pools to drive ductal morphogenesis[44]. In our short term lineage marking experiments (24 h post labelling) we observe that rare Blimp1⁺ cells present in the TEBs rapidly give rise to small clones of Blimp1⁻ luminal cells that clonally expand to generate randomly distributed large luminal clones throughout the proximal to distal axis of the ductal tree during puberty. By contrast, Blimp1⁺ cells widely scattered throughout the adult mammary gland extensively contribute to the maintenance of the virgin ductal epithelium. The percentage of Blimp1⁺ cells remains remarkably low (<0.25%) but relatively constant within pubertal and adult mammary glands, suggesting that Blimp1⁺ cells possess a lower proliferative potential in comparison with their Blimp1⁻ progeny and therefore represent long-lived luminal stem cells.

The Blimp1-expressing cell population that expands in response to pregnancy hormones plays an essential role during maturation of the secretory epithelium. In the absence of Blimp1 milk secretion is severely impaired due to profound defects in the establishment of cell polarity[32]. Here in 3D organoid cultures Blimp1+ cell proliferation was observed in response to Rankl stimulation. The present experiments reveal GFP+ Blimp1-derived progeny within the adult ductal structures are surrounded by clusters of ER+PR+ luminal cells, strongly suggesting that paracrine Rankl signalling rapidly mobilises GFP+ clones to drive alveologenesis at the onset of pregnancy.

Long-lived stem and progenitor cells that survive involution to reform the ductal structures are essential for the regeneration of an extensive secretory alveolar network during repeated pregnancies[4, 5, 8, 11, 13, 45, 46]. Here we observe that clones of Blimp1-derived progeny within the epithelium, initially labelled during puberty, dramatically expand in response to pregnancy-induced hormones giving rise to roughly one-fifth of the alveolar units. Strikingly this contribution remains constant during successive pregnancies and involution. Unfortunately, the multicolour R26R-Confetti reporter allele[47] failed to undergo recombination in combination with our Prdm1CreERT2 allele. Consequently we were unable to determine whether newly formed alveolar structures derive from single or multiple clones.

During involution, when the milk producing epithelium regresses, the majority of surviving GFP+ cells (~58%) expresses Blimp1. This double-positive population, originally labelled during puberty, probably corresponds to the small fraction of Blimp1+ luminal stem cells detectable at all stages examined. Strikingly we failed to find any cells expressing nuclear Blimp1 that lack GFP. Thus our extensive lineage tracing experiments provide no evidence to suggest that Blimp1-expressing cells emerge de novo in the post-natal gland. Rather Blimp1 marks persistent long-lived stem cells that avoid elimination during involution and give rise to ductal and alveolar progenitors during subsequent tissue re-modelling.

An attractive idea is that poorly differentiated breast cancers may arise from luminal progenitors through reactivation of

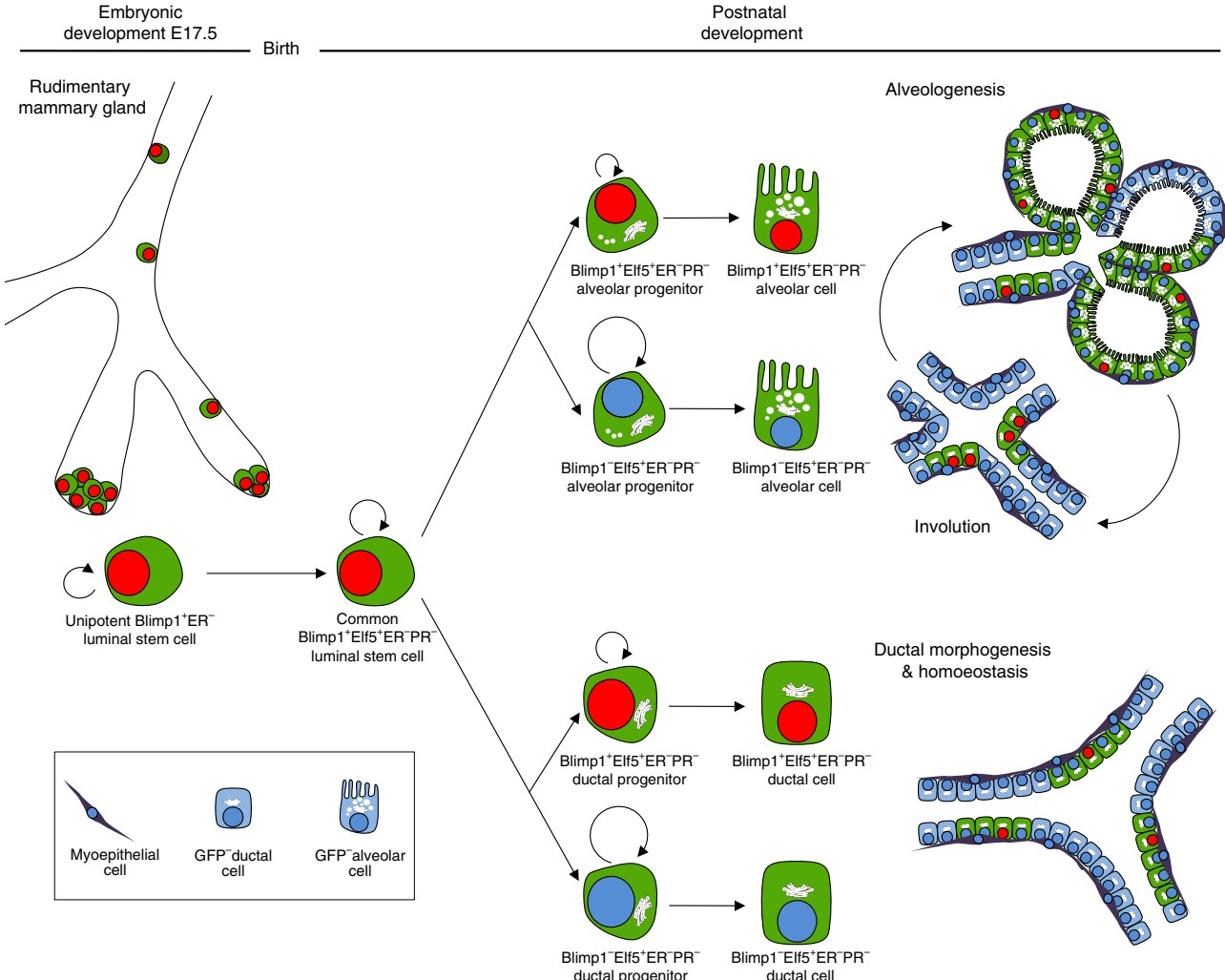

**Fig. 5** Proposed model of Blimp1 mammary cell hierarchy. Lineage tracing during embryonic development uncovers a subset of Blimp1+ unipotent luminal stem cells, which specify around E17.5 where they are committed to the ERα− lineage. These unipotent stem cells contribute exclusively to the luminal compartment postnatally. After birth, a common pool of rare Blimp1+Elf5+ERα−PR− luminal stem cells give rise to both Blimp1+Elf5+ERα−PR− and Blimp1−Elf5+ERα−PR− ductal progenitors, which retain their identity throughout adulthood and contribute extensively to ductal morphogenesis and homoeostasis. In response to pregnancy-induced signals both Blimp1+Elf5+ERα−PR− and Blimp1−Elf5+ERα−PR− alveolar progenitors, emerging from the common Blimp1+Elf5+ERα−PR− luminal stem cell population, are mobilised and clonally expand to contribute extensively to alveologenesis. Blimp1+ cells survive multiple rounds of involution and give rise to Blimp1+ and Blimp1− ductal and alveolar progenitors during subsequent tissue re-modelling, indicating that these cells behave as long-lived stem cells

multipotency[20, 22]. Thus tumours that originate from ERα+ or ERα− luminal progenitors display different growth characteristics, hormonal dependency and treatment outcomes. Previous studies of human breast cancer cell lines demonstrate that Blimp1 regulates EMT and directly represses ERα gene expression[48, 49]. However, this Blimp1 target-binding site is not conserved at the murine ERα gene promoter. Thus the extent to which Blimp1 directly controls ERα expression and possibly governs the balance of ERα+ vs. ERα− cell populations during mammary gland homoeostasis and remodelling, remains unknown. The *Prdm1CreERT2/+* mice represent an ideal model system to specifically generate and follow the fate and behaviour of ERα− luminal progenitors expressing oncogenic mutations. In parallel it will also be highly informative to learn more about Blimp1 functional contributions to tumour formation and heterogeneity in breast cancer model systems.

## Methods

**Animals.** For Blimp1 conditional gene deletion experiments, *K14Cre*[50] mice were crossed with *Prdm1BEH/+* [25] animals to obtain heterozygous *K14Cre;Prdm1BEH/+* males that were then mated with homozygous *Prdm1CA/CA* females[51] to generate control (*Prdm1CA/+* or *K14Cre*) or null (*K14Cre;Prdm1CA/BEH*) females (referred to as K14:Blimp1cKO). Mice (all of C57Bl/6 background) were genotyped by PCR as described in the original reports. For lineage tracing experiments, the *Prdm1CreERT2-LacZ* (hereafter referred to as *Prdm1CreERT2*) allele was generated by introducing a cassette containing a *CreERT2* upstream of IRES-nlacZ followed by a FRT-flanked neo cassette (*CreERT2*-IRES-nLacZ-FRT-neo-FRT) into the ATG translational site in exon 3 via homologous recombination in embryonic stem (ES) cells. Correctly targeted ES cell clones were transiently transfected with a FLP expression construct. Drug-excised subclones were injected into blastocysts to generate germline chimeras. To validate faithful expression of the reporter allele *Prdm1CreERT2/+* embryos were X-gal stained[52].

*Prdm1CreERT2/+* females were intercrossed with homozygous Rosa26-mTmG (referred to as *R26RmTmG*) double-fluorescent reporter males[34], to generate *Prdm1CreERT2/+;R26RmTmG/+* or *R26RmTmG/+* mice. All animal experiments were performed in accordance with Home Office (UK) regulations and approved by the University of Oxford Local Ethical Committee.

**Induction of lineage tracing in vivo.** *Prdm1Cre* activity was induced at 4 or 9 weeks for studies in pubertal and adult virgin *Prdm1CreERT2/+;R26RmTmG/+* female mice, respectively. A single intraperitoneal injection of tamoxifen (TAM; 4 mg/25 g body weight; Sigma-Aldrich) diluted in 100 μl corn oil was administered. This dose of TAM has no noticeable effect on normal mammary epithelial development. Tissues were collected after a 24 h or 72 h chase to determine initial labelling or after 2, 8 or 20 weeks for lineage tracing studies. For lineage tracing experiments during pregnancy and involution, mice were initially induced with TAM at puberty. To induce *Cre* recombination in embryos, pregnant (P) female mice at days P6.5, P7.5, P8.5, P9.5, P11.5, P12.5, P14.5 or P17.5 were injected with a single dose of 0.5 mg/25 g body weight of TAM.

**Whole-mounts and cryosections of mammary glands.** SeeDB-based optical tissue clearing[38] in combination with whole-mount immunolabelling to allow visualization of fluorescent clones in *Prdm1CreERT2/+;R26RmTmG/+* mice was performed on intact tissues. Mammary glands were cut into large pieces (~10 × 10 × 2 mm), fixed in 4% PFA at room temperature (RT) for 2 h, and blocked and permeabilized overnight at 4 °C in PBS with 1% Triton X-100 (Sigma-Aldrich) and 10% bovine serum albumin (BSA; Sigma-Aldrich). Glands were incubated with primary antibodies diluted in blocking buffer at 4 °C for 4 days, and subsequently with secondary antibodies for 2 days at 4 °C, on shaking platform (60 rpm). The following primary antibodies and dilutions were used: GFP (1:1000), K8 (1:200) and K14 (1:1000) (Supplementary Table 1). Secondary antibodies were conjugated to Alexa Fluor 405, Alexa Fluor 488 or Alexa Fluor 647 (Life Technologies), at 1:200 dilutions (Supplementary Table 1). Glands were incubated in 10 mM DAPI (Sigma-Aldrich) for 1 h, then serially incubated for 16 h at RT with gentle rocking in 3 ml of 20, 40, 60 and 80% (w/v) D-(−)-fructose (Sigma-Aldrich) in distilled water containing 0.5% α-thioglycerol (Sigma-Aldrich), and subsequently transferred to 100% (w/v) fructose solution for 24 h followed by 115% (w/v) fructose solution for 2 days or until imaged.

Alternatively, PFA-fixed embryonic and adult mammary gland, embryonic intestine and adult skin frozen sections (20–100 μm thick) were cut and air-dried for 30 min. Following 45 min permeabilization in PBS with 0.2% Triton X-100, slides were blocked for 2 h in 2% BSA, 5% foetal bovine serum (FBS), 0.2% Triton X-100 in PBS. Sections were incubated at 4 °C overnight in primary antibodies, washed and incubated for 1 h at RT with secondary antibodies. The following primary antibodies and dilutions were used: GFP (1:1000), K8 (1:200), K14 (1:1000), Blimp1 (1:100), ERα (1:100), PR (1:100), Elf5 (1:200) and Ki67 (1:200)

(Supplementary Table 1). Secondary antibodies were conjugated to Alexa Fluor 405, Alexa Fluor 488, Alexa Fluor 633 or Alexa Fluor 647 (Life Technologies), at 1:400 dilutions (Supplementary Table 1). Sections were counterstained with DAPI-containing Fluoroshield (Sigma-Aldrich).

**Isolation and culture of primary mammary organoids.** Mammary glands were mechanically dissociated, finely minced and digested for 1 h at 37 °C in DMEM/F-12 medium (Invitrogen) containing 2 mg/ml collagenase A (Roche Diagnostics) and 100 U/ml hyaluronidase (Sigma-Aldrich) on a shaking platform at 140 rpm. Organoids were washed in PBS and separated from single cells through four differential centrifugations (pulse to 1500 rpm in 10 ml DMEM/F12), as previously described[53]. The pellet was then re-suspended in Phenol-free Growth Factor Reduced 100% Matrigel (BD Biosciences), and plated in eight-well coverslip bottom chambers (Fisher Scientific) and kept for 20 min at 37 °C before addition of DMEM-F12 culture media containing Transferrin/Sodium Selenite (Sigma-Aldrich), 5 μg/ml insulin (Sigma-Aldrich) and 1% Penicillin/Streptomycin (Life Technologies). After 24 h, 5 nM 4-OHT (Sigma-Aldrich) was added for 6 h. For testing hormone responses, the medium was supplemented with 200 ng/ml murine Rankl (PeproTech) or 25 ng/ml of progesterone (Sigma-Aldrich) and 50 nM 17β-estradiol (Sigma-Aldrich). Cells were cultured for 7 days.

**Quantitative confocal microscopy and clonal analyses.** Immunofluorescence images were captured with an inverted Olympus FV1200 laser scanning confocal microscope equipped with ×20 dry (UPLSAPO 20X; 0.75 numerical aperture (NA), 0.6 mm working distance (WD)), ×40 silicon oil-immersion (UPLSAPO 40XS; 1.25 NA, 0.3 mm WD) and ×60 oil-immersion (UPLSAPO 60X; 1.40 NA, 0.15 mm WD) Super Apochromat objectives. Z-stacks at 1024 × 1024 pixels were collected at 1 μm intervals. Images were analysed and quantified with ImageJ software.

At least three independent mice were analysed for each developmental time point in vivo. For each traced mammary duct area in whole-mount epithelia, the size and number of GFP+ clones was recorded. The clones were grouped according to size (1 cell, 2–4 cells, >4 cells) and their frequency was calculated vs. the total number of traced clones. We defined clones as single GFP+ cells or clusters of one or more cells that contacted each other. The numbers of GFP+ cells and GFP+ alveoli were determined against the total number of luminal cells or alveoli in traced mammary duct or alveolar areas, respectively. For quantification of GFP, Blimp1 and Ki67 immunofluorescence in 20–100-μm-thick cryosections, 10 regions were randomly chosen per slide at ×40 magnification. The numbers of GFP+ cells were counted against the total number of Blimp1+ or Ki67+ luminal cells in each region. The numbers of Blimp1+ were counted against the total number of GFP+ luminal or alveolar cells in each region.

In 3D organoid assays, three independent experiments were analysed. For each traced organoid, the size and number of GFP+ clones was recorded. The clones were grouped according to size (single cell, clones) and their frequency was calculated vs. the total number of traced clones.

**Isolation of mammary epithelial cells and flow cytometry.** Mammary epithelial cells (MECs) and the separation of basal and luminal cells were performed essentially as described elsewhere[3, 54]. Mechanically dissociated inguinal mammary glands were digested on a shaking platform for 90 min at 37 °C in CO2-independent medium (Invitrogen) containing 5% FBS, 3 mg/ml collagenase A and 100 U/ml hyaluronidase. Cells were pelleted, washed and resuspended in 0.25% trypsin-EDTA (Invitrogen) for 1 min, followed by incubation in 5 mg/ml dispase (Roche Diagnostics) containing 0.1 mg/ml DNase I (Sigma-Aldrich) for 5 min. Contaminating red blood cells were lysed using 0.17 mM NH4Cl. Cells were stained using the following antibodies and dilutions: CD24-PerCP-Cy5.5 (1:200), CD49f-PE-Cy7 (1:50), CD45-APC (1:200) and CD31-APC (1:200) (Supplementary Table 1). DAPI was used to exclude nonviable cells. Basal (CD24low/CD49fhigh) and luminal (CD24high/CD49flow) cells were purified using MoFlo Astrios EQ Flow Cytometer (Beckman Coulter). Data were analysed with FlowJo software.

**Mammary colony-forming assays.** Freshly isolated single mammary cell suspensions were prepared from the inguinal glands. A number of 3000 unsorted cells or sorted GFP+ luminal cells were seeded in 50 μl Matrigel and cultured in DMEM/F12 medium containing 1 mM glutamine, 5% FBS (HyClone Laboratories) and B27 (Life Technologies). Images were captured and colony number and size were analysed with ImageJ software. The cut-off for colony size was 2000 μm².

**Statistical analysis.** Most of the experiments were repeated at least three times and the exact *n* is stated in the corresponding figure legend. GraphPad Prism 7.0 software was used for statistical analysis. Data are shown as mean ± standard error of the mean (s.e.m.). Multiple groups were tested using analysis of variance (ANOVA) with post hoc Tukey test, and comparisons between two groups were performed using *t*-tests. $P < 0.05$ was considered statistically significant. Asterisks indicate levels of significance (*$P < 0.05$; **$P < 0.01$; ***$P < 0.001$). Technically failed 3D cultures or colony-forming assays in which controls did not grow were excluded from the statistical analysis. No animals were excluded from the analyses. Sample sizes were chosen on the basis of our current and previous experiments; no statistical method was applied to predetermine sample size. The experiments were

not randomized. The investigators were not blinded to allocation during experiments and outcome assessment.

**Data availability**. The authors declare that all data supporting the findings of this study are available within the article or from the corresponding author upon reasonable request.

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

## Acknowledgements

We thank Michal Maj and Alan Wainman for valuable assistance with flow cytometry and confocal microscopy, respectively; Jane Rose and Thomas Clague for mouse husbandry; and Xin Lu (University of Oxford) for providing the *K14-Cre* mouse strain. This work was funded by a grant from the Wellcome Trust [WT 102811]. E.J.R. is a Wellcome Trust Principal Research Fellow.

## Author contributions

S.E. and E.J.R. conceived and designed the project. M.A.M., E.J.R. and E.K.B. engineered the *Prdm1Cre^ERT2* mouse strain, and performed Lac-Z staining in embryos. S.E. performed all lineage-tracing experiments. S.E. and E.J.R. analysed and interpreted the data. S.E., E.K.B. and E.J.R. wrote the manuscript. All authors provided intellectual input and approved the final manuscript.

## Additional information

**Competing interests:** The authors declare no competing financial interests.

