## [Peer Review File · Nature Communications]

Reviewers' Comments:

Reviewer #1:

Remarks to the Author:

An earlier study from this team has identified a subpopulation of luminal progenitors expressing Blimp1. These authors have reported that the Blimp1+ luminal cell subset, was highly clonogenic and, similar to already described luminal progenitors, consisted of Elf5+ and ER/PR- cells. Moreover, it was required for alveolar development and differentiation. This new manuscript describes tracing of Blimp1-positive cells throughout mammary development including embryonic stage, ductal and alveolar morphogenesis and involution. The study shows that Blimp1+ cells produce numerous Blimp1-negative cells restricted to luminal ER/PR-negative lineage and serving to build ducts and alveoli. Moreover, these rare Blimp1+ cells are long-lived and retain their identity after several lactation-involution cycles permitting the authors to conclude that Blimp1 expression marks luminal-restricted stem cells.

The study is well designed, the paper is very well written, and in my opinion, this new manuscript represents a significant advance beyond previous publication (Ahmed et al., 2016, Development). Please, find below a few suggestions that could help to more quantitatively evaluate the contribution of Blimp1+ population to the total luminal progenitor pool.

Fig. 1.

Could the authors, in addition to tracing Blimp1-expressing cells, perform immunolabelling for Blimp1 at E18.5? This would show whether Blimp1+ cells proliferate or only give Blimp1- negative progeny at this developmental stage.

Fig. 1b, right lower panel (light green frame). The images do not illustrate well the notion that K14+ (blue) cells stain negative for Blimp1 expression (green).

Figures 2 and 3.

What is the percentage of Blimp1-expressing cells in pubertal and young virgin glands? Further, what percentage of Blimp1-expressing cells responded to Tamoxifen, i.e., switched on GFP? It is not clear whether the authors consider that all Blimp1+ cells were induced.

What is the percentage of Blimp1+ cell progeny (GFP+ cells) in the total luminal population at 12 and 24w of age for the experiments shown in Fig. 2 (scheme Fig.2c) and at 17w for those shown in Fig. 3 (scheme Fig. 3a)?

Could the authors comment on whether Blimp1-expressing luminal stem cells are the only luminal stem cell population within ER/PR-negative lineage?

Elf5+/GFP- cells could originate from Blimp1+ cells that were not induced by Tamoxifen.

Fig. 3g and h. What is the portion of GFP+ clonogenic cells in the total clonogenic luminal population?

Reviewer #2:

Remarks to the Author:

This manuscript presents a very nice lineage tracing study of Blimp-1 expressing stem/progenitor cells and their progeny throughout mammary gland development. Blimp-1 is a zinc finger transcriptional repressor that has been shown to have a role in cell fate decisions in both the embryo and adult in a range of different tissues including the intestine. These authors have previously demonstrated, using gene knockout, that Blimp1 is required for maturation of lactational alveoli. In this new manuscript, the authors have taken a different approach and used the Blimp1 promoter (prdm1) to drive expression of a tamoxifen-inducible Cre recombinase in concert with a floxed STOP switchable red/green fluorescent reporter. Given the current

controversies in the field of mammary stem cell biology where there are opposing views as to the nature/presence of bipotent/unipotent stem cells, this manuscript presents a valuable and interesting contribution.

The work is well carried out, the reporter validated, and the whole mount images are very clear. The only concern is the necessity to use tamoxifen as this has been shown have an impact on mammary gland development. However, with this caveat in mind, we can assume that the lineage tracing has not been perturbed at such low doses.

The first interesting result is that the 'marked' cells in the embryonic mammary gland are all luminal and mark a unipotent lineage that lacks expression of the estrogen receptor alpha. Next, single cell lineage tracing was performed in pubertal females. The authors should provide more evidence that they are indeed labeling a single stem/progenitor cell. This is important if they wish to claim that Blimp1-expressing cells give rise to both ductal morphogenesis and homeostasis.

Have sufficient numbers of clones been examined to conclude conclusively that Blimp1 is never expressed in cells that have been derived from Blimp1 non-expressing progenitors?

The correlation of Blimp1 expressing progenitors with Elf5 expression is interesting. In the summary diagram in Figure 5 it is suggested that expression of Elf5 and ER/PR are mutually exclusive. There is evidence that this is not the case and further immunostaining for Stat5 and Gata3 should be carried out. If the Blimp1+/Elf5+/ER-/PR- cell is a common progenitor for both ductal and alveolar luminal cells, would the authors speculate on the mechanism that favors proliferation of this progenitor during pregnancy as the expense of the ER+/PR+/Gata3+ lineage?

Finally, this is a very interesting manuscript. I think it lacks molecular detail with regard to the roll of Blimp1 and I think it is essential to carry out RNASeq of Blimp1 progeny. These cells can be readily obtained by FACS.

Reviewer #3:

Remarks to the Author:

Mammary gland dynamically changes its morphology through the pregnancy, lactation and involution cycle, and reconstitutes in the next cycle. Although its simple architecture comprises two types of cells, luminal epithelial cells and basal myoepithelial cells, it is not fully understood how the mammary gland maintains the ability of reconstitution through the cycle or which molecule(s) is key to life-time homeostasis of the gland.

The manuscript entitled "Long-lived unipotent Blimp1-positive luminal stem cells drive mammary gland organogenesis throughout adult life" by S. Elias, et al, describes the contribution of Blimp1-positive cells in mammary gland organogenesis. The authors employed Prdm1.CreERT2-LacZ mice crossed with Rosa26mTmG reporter strain, and traced Blimp1-positive cells during development. A key finding of this in vivo study is the ability of Blimp1-positive cells to produce a small subset of Blimp1+/Elf5+/ER- proliferating luminal cells and a large subset of Blimp1-/Elf5-/ER+ differentiated luminal cells; it is potentially crucial to explain how Blimp1 is involved in mammary gland homeostasis. However, several points remain to be resolved.

Major points

1)The core message in this study overlaps with that of the authors' previous paper published in Development (2016; 143, 1663-1673), which showed the requirement of Blimp1+ proliferating luminal cells for mammary gland development and homeostasis. Indeed, Fig.1f, 1g and 3g of this study mimic supplementary Fig.S1 and Fig.2e, 2f of the Development paper, respectively. The authors need to clarify the novelty of this study.

2)The authors successfully described that Blimp1+ luminal cells are a progenitor of Blimp1-/Elf5-/ER+ differentiated luminal cells; however, it is not fully examined whether all of differentiated

luminal cells are produced from Blimp1⁺ cells. Related to this point, the authors did not show what percentage of CD24^{high}/CD49Flow⁺/Krt8⁺ luminal cells was GFP-positive in Fig. 2 through Fig.4. These data would help to assess the indispensability of Blimp1 in mammary gland development and homeostasis.

3) In Fig.3e and 3f, the majority of Ki-67⁺ cells were GFP-negative, while colony-formation efficiency was higher in GFP⁺ cells in Fig. 3g and 3h. How is this discrepancy explained?

4) Blimp1 functions as a transcriptional repressor. However, the expression of downstream Blimp1 targets was not analyzed in this study. The authors need to examine some molecular role(s) of Blimp1 for mammary gland development and homeostasis.

Minor points

1) In Fig. 1b, labelling with Krt8 and Krt14 is missing.

2) In manuscript page 8, line 1-2, the figure numbers are missing.

3) In Fig.2d, 2e, 3f, 4c, 4d and 4f, what types of cells were composed of the population is not mentioned.

RESPONSE TO REVIEWERS COMMENTS

Reviewers' comments:

Reviewer #1 (Remarks to the Author):

An earlier study from this team has identified a subpopulation of luminal progenitors expressing Blimp1. These authors have reported that the Blimp1+ luminal cell subset, was highly clonogenic and, similar to already described luminal progenitors, consisted of Elf5+ and ER/PR- cells. Moreover, it was required for alveolar development and differentiation. This new manuscript describes tracing of Blimp1-positive cells throughout mammary development including embryonic stage, ductal and alveolar morphogenesis and involution. The study shows that Blimp1+ cells produce numerous Blimp1-negative cells restricted to luminal ER/PR-negative lineage and serving to build ducts and alveoli. Moreover, these rare Blimp1+ cells are long-lived and retain their identity after several lactation-involution cycles permitting the authors to conclude that Blimp1 expression marks luminal-restricted stem cells.

The study is well designed, the paper is very well written, and in my opinion, this new manuscript represents a significant advance beyond previous publication (Ahmed et al., 2016, Development). Please, find below a few suggestions that could help to more quantitatively evaluate the contribution of Blimp1+ population to the total luminal progenitor pool.

Fig. 1.

Could the authors, in addition to tracing Blimp1-expressing cells, perform immunolabelling for Blimp1 at E18.5? This would show whether Blimp1+ cells proliferate or only give Blimp1- negative progeny at this developmental stage.

- To assess whether Blimp1 positive cells self-renew and/or give rise to Blimp1-negative progeny we now include a representative confocal image of Blimp1 staining at E18.5 in Panel 1d and provide a quantitation of the ratio of Blimp1+/GFP- versus Blimp1+ GFP+ showing that the majority (>75%) of GFP+ cells express Blimp1. These new results are described in the text.

Fig. 1b, right lower panel (light green frame). The images do not illustrate well the notion that K14+ (blue) cells stain negative for Blimp1 expression (green).

- To more clearly establish this point we have amended Panel 1c to include line scan quantifications using ImageJ software showing that GFP+ cells co-localise with K8, but not K14.

Figures 2 and 3.

What is the percentage of Blimp1-expressing cells in pubertal and young virgin glands? Further, what percentage of Blimp1-expressing cells responded to Tamoxifen, i.e., switched on GFP? It is not clear whether the authors consider that all Blimp1+ cells were induced.

- Both the other reviewers also asked us for more quantitative data (see below). The revised Results section (page 7) explains that Blimp1-expressing cells (identified by activation of GFP-expression at 24h post tamoxifen treatment) represent $0.25 \pm 0.05\%$ (puberty) and $0.2 \pm 0.03\%$ (adult virgin) of the total luminal population. We have also amended both Fig.2e and Supplementary Fig.2e to show a time course of the numbers of GFP+ cells detected post TAM-administration.

- Importantly our pilot titration experiments established that the low dose of TAM used here (4mg/25g body weight) maximally labelled the Blimp1CreERT2+ luminal cells and that 24hrs post-injection the representation of GFP+ cells precisely mirrors those expressing the Blimp1 VenusBAC transgene positive cells (Ohinata et al, Nature 2008). The revised Figures now include additional quantitation showing that all luminal Blimp1+ ($99 \pm 0.8\%$) switch on mGFP after 24h of tamoxifen induction (Fig.2g: puberty; Supplementary Fig.2g: adult virgin).

Moreover none of the Blimp1+ cells were entirely devoid of GFP. The exceptional Blimp1+GFP- cells (~1% of total Blimp1+ population) at 24h post-TAM administration express cytosolic GFP. This population probably represents the small sub-set where GFP protein has not yet trafficked to the plasma membrane. Indeed at later time points all Blimp1+ progeny expressed membrane-bound GFP. Moreover as shown in Fig. 4e, 100% of Blimp1+ cells that survive involution (identified by staining for endogenous Blimp1 nuclear protein) are GFP+. These findings rule out any de novo Blimp1 expression by luminal cells in the post-natal gland and demonstrate conclusively that all Blimp1+ cells were induced to express GFP under these labelling conditions.

What is the percentage of Blimp1+ cell progeny (GFP+ cells) in the total luminal population at 12 and 24w of age for the experiments shown in Fig. 2 (scheme Fig.2c) and at 17w for those shown in Fig. 3 (scheme Fig. 3a)?

- To address this point also raised by Reviewer #3, we have added quantification of the percentage GFP+ cells within the total luminal population (new Fig.2e; new Supplementary Fig.2e).

Could the authors comment on whether Blimp1-expressing luminal stem cells are the only luminal stem cell population within ER/PR-negative lineage?

- Results shown in Fig. 3d demonstrate that all GFP+ cells are ER-PR-, and are surrounded by GFP-ER+PR+ cell clusters, as well as numerous GFP-ER-PR- cells 24h post-Tam. However by 8 weeks later, clonal expansion of GFP+ER-PR- cells is accompanied by an apparent reduction in the GFP-ER-PR- cell population, suggesting that Blimp1-expressing luminal stem cells contribute extensively to the ER-PR- lineage. Nonetheless, we still detect GFP-ER-PR- luminal cells, implicating an additional Blimp1-stem cell population(s) within the ER/PR-negative lineage. Consistent with this interpretation the majority of the alveoli formed during pregnancy are generated from the GFP- luminal cells.

Elf5+/GFP- cells could originate from Blimp1+ cells that were not induced by Tamoxifen.

- As discussed above (Point 3), TAM efficiently labels all Blimp1+ luminal stem cells (New Fig.2f,g: puberty; Supplementary Fig.2f,g: adult virgin). Therefore, as for GFP-ER-PR- cells, GFP-Elf5+ cells must arise from a distinct Blimp1- stem cell subset(s). Similarly the recent work from the Blanpain lab (Van Keymeulen et al., 2017) showed that ER+ luminal stem cells are lineage restricted and do not give rise to ER- luminal progeny. The amended Discussion highlights these points. However, further studies will be required to clarify functional hierarchies.

Fig. 3g and h. What is the portion of GFP+ clonogenic cells in the total clonogenic luminal population?

- Quantification showing that the portion of GFP+ clonogenic cells within the total luminal population is $59 \pm 2.4\%$, has been added to the Results.

--

Reviewer #2 (Remarks to the Author):

This manuscript presents a very nice lineage tracing study of Blimp-1 expressing stem/progenitor cells and their progeny throughout mammary gland development. Blimp-1 is a zinc finger transcriptional repressor that has been shown to have a role in cell fate decisions in both the embryo and adult in a range of different tissues including the intestine. These authors have previously demonstrated, using gene knockout, that Blimp1 is required for maturation of lactational alveoli. In this new manuscript, the authors have taken a different approach and used the Blimp1 promoter (*prdm1*) to drive expression of a tamoxifen-inducible Cre recombinase in concert with a floxed STOP switchable red/green fluorescent reporter. Given the current controversies in the field of mammary stem cell biology where there are opposing views as to the nature/presence of bipotent/unipotent stem cells, this manuscript presents a valuable and interesting contribution.

The work is well carried out, the reporter validated, and the whole mount images are very clear. The only concern is the necessity to use tamoxifen as this has been shown to have an impact on mammary gland development. However, with this caveat in mind, we can assume that the lineage tracing has not been perturbed at such low doses.

The first interesting result is that the 'marked' cells in the embryonic mammary gland are all luminal and mark a unipotent lineage that lacks expression of the estrogen receptor alpha. Next, single cell lineage tracing was performed in pubertal females. The authors should provide more evidence that they are indeed labeling a single stem/progenitor cell. This is important if they wish to claim that Blimp1-expressing cells give rise to both ductal morphogenesis and homeostasis.

- We carefully optimised SeeDB-based tissue clearing to enable us to conduct unbiased clonal analyses. Additionally to visualise deep regions at high resolution in whole mount glands, we used an Olympus confocal microscope (FV1200 IX83; Olympus) equipped with a 40x (1.25 numerical aperture (NA), 0.3-mm working distance) silicon oil-immersion Super Apochromatic lens (UPLSAPO 40XS). Compared to conventional oil immersion objectives, this objective significantly reduces loss of contrast due to spherical aberration and provides higher resolution and brightness, especially when

imaging thick samples. Together these technical advances allowed us to easily visualise single GFP cell clones at 24h-post tamoxifen induction. Details of the numbers of clones analysed (abundance, size and distribution) are stated in the figure legends and quantitated in the Figures, and additional quantitative data has been provided (Fig.2e: puberty; Supplementary Fig.2e: adult virgin).

Have sufficient numbers of clones been examined to conclude conclusively that Blimp1 is never expressed in cells that have been derived from Blimp1 non-expressing progenitors?

- To address this point also raised by the other Reviewers, we now include histograms (Fig.2g and Supplementary Fig.2g) showing that 24h-post TAM induction $99 \pm 0.8\%$ of all Blimp1-expressing cells activate GFP. We analysed 3 mice per time point, with 10 random regions being assessed at x40 magnification. Numbers of Blimp1+GFP+ vs Blimp1+GFP- cells were compared to the total number of Blimp1+ cells in each region, using ImageJ software.

As discussed above (Reviewer 1, point 3) the exceptional Blimp1+ cells (~1%) at 24h post-TAM administration express cytoplasmic GFP whereas at later time points ALL Blimp1+ cells expressed membrane-bound GFP. We are confident that our conclusions that the Blimp1+ stem cells specified in the late stage embryonic gland are indeed long-lived, and Blimp1 is not re-expressed de novo in luminal cells during later stages of morphogenesis, homeostasis or regeneration following involution (Fig. 4) are well justified.

The correlation of Blimp1 expressing progenitors with Elf5 expression is interesting. In the summary diagram in Figure 5 it is suggested that expression of Elf5 and ER/PR are mutually exclusive. There is evidence that this is not the case and further immunostaining for Stat5 and Gata3 should be carried out. If the Blimp1+/Elf5+/ER-/PR- cell is a common progenitor for both ductal and alveolar luminal cells, would the authors speculate on the mechanism that favors proliferation of this progenitor during pregnancy as the expense of the ER+/PR+/Gata3+ lineage?

- The complex functional relationships between Elf5+/ER-/PR- and ER+/PR+/Gata3+ cell populations remains ill defined. It is well known that Gata3 plays an essential role in luminal differentiation and marks ER+ cells which display very low proliferative rates (Kouros-Mehr et al. 2006; Cell 127, 1041–1055.). Consistent with this we find in adult virgin mice, at 24h-post Tam, Gata3 expression is restricted to GFP- cells. 8 weeks later the majority of GFP+ cells still lack Gata3 expression, but surprisingly we find a minority (~13%) weakly expressing Gata3 (see Figure attached). Remarkably, when we stained cells for pStat5 and Gata3 at mid-pregnancy, we observed a large fraction (~62%) of GFP+ cells are pStat5+, while only ~31% of GFP+ co-express Gata3+.

A recent study (Oliver et al., 2012; Genes & Development 26:1086–1097) reported a population of luminal progenitors co-expressing Gata3 and p-Stat5, where Gata3 represses the transcriptional repressor Zfp15. Interestingly, this regulatory function of Gata3, while crucial for alveologenesis, promotes the expansion of proliferative pStat5 single-positive cells at the expense of the double-positive Gata-3+/pStat5 cells, which become a very minor population (Oliver et al., Fig 3.). These observations further highlight the dynamic expression of Gata3 and pStat5, but also suggest that Gata3 may

not be essential for the differentiation or maintenance of alveolar cells. We speculate that the non-proliferative Gata3+ER+PR+ cells signal in a paracrine fashion (probably through RANK ligand as shown in Supplementary Fig. 5), to promote Blimp1+Elf5+pStat5+ER-PR- alveolar cell proliferation and differentiation.

Our preliminary staining and quantitation data is shown below. We agree that it will be interesting to dissect the functional relationship between Blimp1, Gata3 and pStat5 in future studies. However in the absence of more detailed experiments to provide mechanistic insights further speculation is unwarranted.

Expression of Gata3 and p-Stat5a by Blimp1-progeny during development.

(a) Illustration of the lineage tracing strategy adopted in **(b,c)**. 9-week-old adult virgin *Prdm1Cre^{ERT2/+};R26R^{mTmG/+}* females injected with TAM (4 mg/25 g body weight) were analysed 24h or 8w later (24h or 8w p.i.). **(b)** Cryosections (50 μ m-thick) from adult virgin mammary glands analysed at 24h and 8w post-TAM induction (p.i.) stained for GFP (green) and Gata3 (red) and counterstained with DAPI (Nuclei, blue). All GFP labelled cells at 24h post-TAM induction are Gata3⁻, whereas at 8w post-TAM induction, both GFP⁺Gata3⁻ cell clusters and GFP⁺Gata3⁺ individual cells (arrow) are observed ($n=3$ mice per time point). Scale bars, 50 μ m. **(c)** Histograms showing percentages in adulthood of GFP⁺Gata3⁺ versus GFP⁺Gata3⁻ cells at 24h and 20w post-TAM induction in adulthood. Data are presented as mean \pm s.e.m. (Number of GFP⁺ cells analysed pooled from $n=3$ mice per time point: $n=154$ and $n=681$, respectively). *** $P<0.001$ (t test). **(d)** Illustration of the lineage tracing strategy adopted in **(e,f)**. 4-week-old adult virgin *Prdm1Cre^{ERT2/+};R26R^{mTmG/+}* females were injected with TAM (4 mg/25 g body weight) and analysed at day 14.5 of the first pregnancy (P1). **(e)** Cryosections (20 μ m-thick) from P1, mammary glands stained for GFP (green) and Gata3 or pStat5a (red) and counterstained with DAPI (Nuclei, blue). The proportion of GFP⁺Gata3⁺ is lower than that of GFP⁺pStat5a⁺ cells ($n=3$ mice per time point). Scale bars, 20 μ m. **(f)** Histograms showing percentages of GFP⁺Gata3⁺ versus GFP⁺pStat5a⁺ cells at P1. Data are presented as mean \pm s.e.m. (Number of analysed GFP⁺ cells pooled from $n=3$ mice per time point: $n=1736$ and $n=1562$, respectively). *** $P<0.001$ (t test).

Finally, this is a very interesting manuscript. I think it lacks molecular detail with regard to the roll of Blimp1 and I think it is essential to carry out RNASeq of Blimp1 progeny.

- As noted by the Reviewers, there exists extensive heterogeneity within the luminal progenitor compartment, specifically with respect to expression of key markers such as Elf5, ER α , PR (not to mention Prominin-1, Sox9, Lgr6, Notch etc). Molecular characterization of these sub-types has yet to be reported. RNA-Seq analysis of rare GFP+ive Blimp1 progeny in comparison with heterogeneous GFP negative luminal cells would not help clarify these relationships and would be inadequate to define a unique cell type. Moreover, the molecular profile of GFP+ progeny probably changes during prepubertal, pubertal, virgin, pregnancy stages and according to their position within the mammary tree.

Single cell RNA-seq analysis could potentially provide new information about luminal cell progenitors and possibly unique transcriptomic features of the Blimp1+ stem cells and their Blimp1-progeny at different time points of organogenesis. However based on our previous single-cell analyses (Nelson et al, Nature Comms, 2016 DOI: 10.1038/ncomms11414) this work would require a challenging 2 years plus experimental time frame at best. Importantly however, aside from Reviewer 2, none of the other reviewers have requested RNA-Seq analysis.

These cells can be readily obtained by FACS.

--

Reviewer #3 (Remarks to the Author):

Mammary gland dynamically changes its morphology through the pregnancy, lactation and involution cycle, and reconstitutes in the next cycle. Although its simple architecture comprises two types of cells, luminal epithelial cells and basal myoepithelial cells, it is not fully understood how the mammary gland maintains the ability of reconstitution through the cycle or which molecule(s) is key to life-time homeostasis of the gland. The manuscript entitled "Long-lived unipotent Blimp1-positive luminal stem cells drive mammary gland organogenesis throughout adult life" by S. Elias, et al, describes the contribution of Blimp1-positive cells in mammary gland organogenesis. The authors employed Prdm1.CreERT2-LacZ mice crossed with Rosa26mTmG reporter strain, and traced Blimp1-positive cells during development. A key finding of this in vivo study is the ability of Blimp1-positive cells to produce a small subset of Blimp1+/Elf5+/ER- proliferating luminal cells and a large subset of Blimp1-/Elf5-/ER+ differentiated luminal cells; it is potentially crucial to explain how Blimp1 is involved in mammary gland homeostasis. However, several points remain to be resolved.

Major points

1) The core message in this study overlaps with that of the authors' previous paper published in *Development* (2016;143, 1663-1673), which showed the requirement of Blimp1+ proliferating luminal cells for mammary gland development and homeostasis. Indeed, Fig.1f, 1g and 3g of this study mimic supplementary Fig.S1 and Fig.2e, 2f of the *Development* paper, respectively. The authors need to clarify the novelty of this study.

- Our previous paper exploited a VenusBAC Blimp1-reporter transgene to identify a discrete sub-set of highly clonogenic luminal progenitor cells, and characterized the loss

of function phenotype (Ahmed et al., 2016, *Development*, 143, 1663-1673). However we were unable to describe the normal behaviour and characteristics of Blimp1+ cells in the undisturbed epithelium, and could not evaluate whether Blimp1 expression in luminal cells fluctuates stochastically. In contrast here we generated a novel Blimp1/Prdm1.CreERT2 knock-in allele strain and performed lineage tracing of this rare luminal cell population via 3D whole tissue imaging at discrete stages of mammary gland morphogenesis and homeostasis. Our analysis provides conclusive evidence for the existence of a long-lived unipotent luminal stem cell population specified at late embryonic stages that survives multiple rounds of involution, and contributes to mammary gland growth, remodelling and homeostasis throughout adult life.

2)The authors successfully described that Blimp1+ luminal cells are a progenitor of Blimp1-/Elf5-/ER+ differentiated luminal cells; however, it is not fully examined whether all of differentiated luminal cells are produced from Blimp1+ cells. Related to this point, the authors did not show what percentage of CD24^{high}/CD49^{low}/Krt8+ luminal cells was GFP-positive in Fig. 2 through Fig.4. These data would help to assess the indispensability of Blimp1 in mammary gland development and homeostasis.

- To address this point, raised by both other Reviewers, we have included further quantitation (Fig.2e: puberty; Supplementary Fig.2e: adult virgin). We also added quantifications showing the percentage GFP+ cells within the total luminal population. However as discussed above, considerable evidence suggests that Blimp1- stem cells also contribute to epithelial morphogenesis and homeostasis.

3)In Fig.3e and 3f, the majority of Ki-67+ cells were GFP-negative, while colony-formation efficiency was higher in GFP+ cells in Fig. 3g and 3h. How is this discrepancy explained?

- Yes, the majority of Ki67+ cells are GFP- at 24h-post tamoxifen induction. However as shown in Fig. 3e 8 weeks later the number of Ki67+GFP- cells appears to decrease. We have now amended Fig. 3f to better reflect the proportions of Ki67+GFP+ versus Ki67+GFP-, and thus the expansion of this highly clonogenic GFP+Ki67+ cells at the expense of GFP-Ki67+ population.

4)Blimp1 functions as a transcriptional repressor. However, the expression of downstream Blimp1 targets was not analyzed in this study. The authors need to examine some molecular role(s) of Blimp1 for mammary gland development and homeostasis.

- Here we exploit Blimp1 as a marker of an extremely rare stem cell population. Indeed the Blimp1-expressing cell population constitutes only ~0.2% of the total luminal population. Blimp-1 ChIP Seq analysis previously reported by ourselves and others exploited GFP-tagged knock-in alleles in combination with proven ChIP-grade anti-GFP mAb (Mould et al, 2015; Magnusdottir et al., 2013). Genome-wide high throughput methodologies would be technically impossible here.

Minor points

1)In Fig.1b, labelling with Krt8 and Krt14 is missing.

2)In manuscript page 8, line 1-2, the figure numbers are missing.

3)In Fig.2d, 2e, 3f, 4c, 4d and 4f, what types of cells were composed of the population is not mentioned.

- We thank the reviewer for pointing out these omissions. The errors have all been corrected, and missing Figure numbers inserted into the text.

Reviewers' Comments:

Reviewer #1:

Remarks to the Author:

All critical comments have been addressed perfectly.

Reviewer #2:

Remarks to the Author:

The authors have improved the manuscript and have addressed my concerns in a satisfactory manner.

Reviewer #3:

Remarks to the Author:

Reviewer #3 (Remarks to the Author):

Mammary gland dynamically changes its morphology through the pregnancy, lactation and involution cycle, and reconstitutes in the next cycle. Although its simple architecture comprises two types of cells, luminal epithelial cells and basal myoepithelial cells, it is not fully understood how the mammary gland maintains the ability of reconstitution through the cycle or which molecule(s) is key to life-time homeostasis of the gland. The manuscript entitled "Long-lived unipotent Blimp1-positive luminal stem cells drive mammary gland organogenesis throughout adult life" by S. Elias, et al, describes the contribution of Blimp1-positive cells in mammary gland organogenesis. The authors employed Prdm1.CreERT2-LacZ mice crossed with Rosa26mTmG reporter strain, and traced Blimp1-positive cells during development. A key finding of this in vivo study is the ability of Blimp1-positive cells to produce a small subset of Blimp1+/Elf5+/ERproliferating luminal cells and a large subset of Blimp1-/Elf5-/ER+ differentiated luminal cells; it is potentially crucial to explain how Blimp1 is involved in mammary gland homeostasis. However, several points remain to be resolved.

Major points 1) The core message in this study overlaps with that of the authors' previous paper published in Development (2016; 143, 1663-1673), which showed the requirement of Blimp1+ proliferating luminal cells for mammary gland development and homeostasis. Indeed, Fig.1f, 1g and 3g of this study mimic supplementary Fig.S1 and Fig.2e, 2f of the Development paper, respectively. The authors need to clarify the novelty of this study.

- Our previous paper exploited a VenusBAC Blimp1-reporter transgene to identify a discrete subset of highly clonogenic luminal progenitor cells, and characterized the loss of function phenotype (Ahmed et al., 2016, Development, 143, 1663-1673). However we were unable to describe the normal behaviour and characteristics of Blimp1+ cells in the undisturbed epithelium, and could not evaluate whether Blimp1 expression in luminal cells fluctuates stochastically. In contrast here we generated a novel Blimp1/Prdm1.CreERT2 knock-in allele strain and performed lineage tracing of this rare luminal cell population via 3D whole tissue imaging at discrete stages of mammary gland morphogenesis and homeostasis. Our analysis provides conclusive evidence for the existence of a long-lived unipotent luminal stem cell population specified at late embryonic stages that survives multiple rounds of involution, and contributes to mammary gland growth, remodelling and homeostasis throughout adult life.

I accept the authors' explanation.

2) The authors successfully described that Blimp1+ luminal cells are a progenitor of Blimp1-/Elf5-/ER+ differentiated luminal cells; however, it is not fully examined whether all of differentiated luminal cells are produced from Blimp1+ cells. Related to this point, the authors did not show what

percentage of CD24^{high}/CD49^{low}/Krt8⁺ luminal cells was GFP-positive in Fig. 2 through Fig.4. These data would help to assess the indispensability of Blimp1 in mammary gland development and homeostasis.

- To address this point, raised by both other Reviewers, we have included further quantitation (Fig.2e: puberty; Supplementary Fig.2e: adult virgin). We also added quantifications showing the percentage GFP⁺ cells within the total luminal population. However as discussed above, considerable evidence suggests that Blimp1⁻ stem cells also contribute to epithelial morphogenesis and homeostasis.

A key point of this manuscript is the Blimp1⁺ cells are progenitors of the luminal cells. However, quantitative evidence is still missing. For instance, in Fig.2f (right panel) almost all ductal cells seem GFP⁺, supporting the authors' conclusion. But the authors need to quantify this by demonstrating how many cells were GFP⁺ in all of luminal cells (NOT IN BLIMP1⁺ CELLS) in Fig.2g. Alternatively, they might need to show how many cells were GFP⁺ in a gated population of CD24^{high}/CD49^{low} luminal cells. Similar quantitations should be done in other figures too. Such quantitative evidence is necessary to show how important Blimp1⁺ cells are as a stem for luminal progenitors.

Also, some sentences and figures are confusing to me: for instance, though 100% of GFP⁺ cells were Blimp1⁺ in Fig. 2g, it does not seem to me that it was so in Fig. 2f.

3) In Fig.3e and 3f, the majority of Ki-67⁺ cells were GFP-negative, while colony formation efficiency was higher in GFP⁺ cells in Fig. 3g and 3h. How is this discrepancy explained?

- Yes, the majority of Ki67⁺ cells are GFP⁻ at 24h-post tamoxifen induction. However as shown in Fig. 3e 8 weeks later the number of Ki67⁺GFP⁻ cells appears to decrease. We have now amended Fig. 3f to better reflect the proportions of Ki67⁺GFP⁺ versus Ki67⁺GFP⁻, and thus the expansion of this highly clonogenic GFP⁺Ki67⁺ cells at the expense of GFP⁻Ki67⁺ population.

From Fig. 3f and 3g-h, I understand that GFP⁺ cells were potentially proliferative in the colony-formation assay. But even when the colony-forming cells were GFP⁺, as far as I know, it would be impossible to examine whether they were Ki-67⁺ or not. So, the explanation the authors gave is not enough. In addition, I should ask if GFP-negative cells used for the colony-formation assay contained neither basal nor stromal cells in Fig. 3g-h.

4) Blimp1 functions as a transcriptional repressor. However, the expression of downstream Blimp1 targets was not analyzed in this study. The authors need to examine some molecular role(s) of Blimp1 for mammary gland development and homeostasis.

- Here we exploit Blimp1 as a marker of an extremely rare stem cell population. Indeed the Blimp1-expressing cell population constitutes only ~0.2% of the total luminal population. Blimp-1 ChIP Seq analysis previously reported by ourselves and others exploited GFP-tagged knock-in alleles in combination with proven ChIP-grade anti-GFP mAb (Mould et al, 2015; Magnusdottir et al., 2013). Genome-wide high throughput methodologies would be technically impossible here.

Examining whether Blimp1 overexpression suppresses Elf5 (or ER/PgR) expression in cultured cells would support the authors' notion of Blimp1 in this manuscript.

Minor points 1) In Fig.1b, labelling with Krt8 and Krt14 is missing. 2) In manuscript page 8, line 1-2, the figure numbers are missing. 3) In Fig.2d, 2e, 3f, 4c, 4d and 4f, what types of cells were composed of the population is not mentioned.

- We thank the reviewer for pointing out these omissions. The errors have all been corrected, and missing Figure numbers inserted into the text.

Please make sure that some numbers in the manuscript do not show in the figures: for instance, where do "~59% of total" (page 9, line 214) come from?

REVIEWERS' COMMENTS:

Reviewer #1 (Remarks to the Author):

All critical comments have been addressed perfectly.

--

Reviewer #2 (Remarks to the Author):

The authors have improved the manuscript and have addressed my concerns in a satisfactory manner.

--

Reviewer #3 (Remarks to the Author):

Mammary gland dynamically changes its morphology through the pregnancy, lactation and involution cycle, and reconstitutes in the next cycle. Although its simple architecture comprises two types of cells, luminal epithelial cells and basal myoepithelial cells, it is not fully understood how the mammary gland maintains the ability of reconstitution through the cycle or which molecule(s) is key to life-time homeostasis of the gland. The manuscript entitled "Long-lived unipotent Blimp1-positive luminal stem cells drive mammary gland organogenesis throughout adult life" by S. Elias, et al, describes the contribution of Blimp1-positive cells in mammary gland organogenesis. The authors employed Prdm1.CreERT2-LacZ mice crossed with Rosa26mTmG reporter strain, and traced Blimp1-positive cells during development. A key finding of this in vivo study is the ability of Blimp1-positive cells to produce a small subset of Blimp1+/Elf5+/ERproliferating luminal cells and a large subset of Blimp1-/Elf5-/ER+ differentiated luminal cells; it is potentially crucial to explain how Blimp1 is involved in mammary gland homeostasis. However, several points remain to be resolved.

Major points 1)The core message in this study overlaps with that of the authors' previous paper published in Development (2016;143, 1663-1673) , which showed the requirement of Blimp1+ proliferating luminal cells for mammary gland development and homeostasis. Indeed, Fig.1f, 1g and 3g of this study mimic supplementary Fig.S1 and Fig.2e, 2f of the Development paper, respectively. The authors need to clarify the novelty of this study.

Initial Response: *Our previous paper exploited a VenusBAC Blimp1-reporter transgene to identify a discrete sub-set of highly clonogenic luminal progenitor cells, and characterized the loss of function phenotype (Ahmed et al., 2016, Development, 143, 1663-1673). However we were unable to describe the normal behaviour and characteristics of Blimp1+ cells in the undisturbed epithelium, and could not evaluate whether Blimp1 expression in luminal cells fluctuates stochastically. In contrast here we generated a novel Blimp1/Prdm1.CreERT2 knock-in allele strain and performed lineage tracing of this rare luminal cell population via 3D whole tissue imaging at discrete stages of mammary gland morphogenesis and homeostasis. Our analysis provides conclusive evidence for the existence of a long-lived unipotent luminal stem cell population specified at late embryonic stages that survives multiple rounds of involution, and contributes to mammary gland growth, remodelling and homeostasis throughout adult*

life.

I accept the authors' explanation.

2)The authors successfully described that Blimp1+ luminal cells are a progenitor of Blimp1-/Elf5-/ER+ differentiated luminal cells; however, it is not fully examined whether all of differentiated luminal cells are produced from Blimp1+ cells.

Second Response: *The major finding of our paper is quite the opposite: the rare Blimp1+ cells give rise exclusively to the Elf5+ERa- luminal progenitors. We also showed in the original paper that only a percentage of the luminal cell population is derived from these rare Blimp1+ stem cells.*

Related to this point, the authors did not show what percentage of CD24high/CD49Flow/Krt8+ luminal cells was GFP-positive in Fig. 2 through Fig.4. These data would help to assess the indispensability of Blimp1 in mammary gland development and homeostasis.

Initial Response: *To address this point, raised by both other Reviewers, we have included further quantitation (Fig.2e: puberty; Supplementary Fig.2e: adult virgin). We also added quantifications showing the percentage GFP+ cells within the total luminal population. However as discussed above, considerable evidence suggests that Blimp1- stem cells also contribute to epithelial morphogenesis and homeostasis.*

A key point of this manuscript is the Blimp1+ cells are progenitors of the luminal cells. However, quantitative evidence is still missing. For instance, in Fig.2f (right panel) almost all ductal cells seem GFP+, supporting the authors' conclusion. But the authors need to quantify this by demonstrating how many cells were GFP+ in all of luminal cells (NOT IN BLIMP1+ CELLS) in Fig.2g.

Second Response: *The Reviewer does not appear to have understood the data presented. For clarification. The very rare Blimp1+ cells give rise mostly to GFP+Blimp1- progeny. As requested by the other Reviewers we precisely quantitated the clone sizes and contribution to the luminal cell population with time in Panels 2 d & e. The purpose of the data shown in panel 2f is to show that all of the Blimp1+ cells are also GFP+ when initially labelled and retain the label 20 weeks later. By contrast Panel 2h quantitates the observation that the bulk of the labelled GFP progeny do not retain Blimp1 expression. The data they specifically requested "how many cells were GFP+ in all of luminal cells" is provided in the graph in Panel 2e.*

Alternatively, they might need to show how many cells were GFP+ in a gated population of CD24high/CD49low luminal cells. Similar quantitations should be done in other figures too. Such quantitative evidence is necessary to show how important Blimp1+ cells are as a stem for luminal progenitors.

Second Response: *We have also provided this same quantitative data in Supp Figure 2, which is in complete agreement with that shown in Figure 2. Both Figures show the very rare population of Blimp1+ stem cells gives rise to around 15% of the total number of luminal cells 20 weeks post chase during both ductal morphogenesis and in the resting adult gland. Experiments shown in Figure 4 demonstrate these represent a sub-*

set of the *Elf5+ERa-* luminal progenitors, and give rise to around a fifth of the alveolar units during pregnancy. Throughout the manuscript we have provided extensive and rigorous quantitative data - as explained in the legends and methods we have painstakingly counted many thousands of cells.

Also, some sentences and figures are confusing to me: for instance, though 100% of GFP+ cells were *Blimp1+* in Fig. 2g, it does not seem to me that it was so in Fig. 2f.

Second Response: *In Figure 2g we are making the important point that the *Blimp1+* cells are long lived. 24hours after tamoxifen induction they are GFP+ and retain GFP expression 20weeks later, showing they are long-lived (if they arose de novo, they could not express GFP). Fig 2f makes a completely different point namely that 20 weeks post-labelling the bulk of the GFP+ progeny do not express *Blimp1* (this finding is very carefully quantitated in Panel 2h). Collectively these data provide unambiguous evidence that the *Blimp1+* cells are long-lived stem cells that persist within the luminal compartment and give rise to GFP+*Blimp1-* luminal progenitors through the life of the animal.*

3) In Fig. 3e and 3f, the majority of *Ki-67+* cells were GFP-negative, while colony formation efficiency was higher in GFP+ cells in Fig. 3g and 3h. How is this discrepancy explained?

Initial Response: *Yes, the majority of *Ki67+* cells are GFP- at 24h-post tamoxifen induction. However as shown in Fig. 3e 8 weeks later the number of *Ki67+*GFP- cells appears to decrease. We have now amended Fig. 3f to better reflect the proportions of *Ki67+*GFP+ versus *Ki67+*GFP-, and thus the expansion of this highly clonogenic GFP+*Ki67+* cells at the expense of GFP-*Ki67+* population.*

From Fig. 3f and 3g-h, I understand that GFP+ cells were potentially proliferative in the colony-formation assay. But even when the colony-forming cells were GFP+, as far as I know, it would be impossible to examine whether they were *Ki-67+* or not. So, the explanation the authors gave is not enough.

Second Response: *Colony forming assays are routinely used to test the potential of isolated mammary epithelial cells. We agree that we cannot (and indeed do not) say that the GFP+ colonies in vitro were derived from the *Ki67+* cells tracked in vivo, because these are 2 separate experiments using discrete pools of animals. It is standard practice in the field to perform these complementary experiments: CFA provide an accessible culture system which allows us to assess and compare the behaviour of different cell populations (i.e. GFP+ vs GFP- vs total), and in doing so we reinforce our in vivo lineage data that show that GFP+ cells are endowed with a high clonogenic capacity.*

In addition, I should ask if GFP-negative cells used for the colony-formation assay contained neither basal nor stromal cells in Fig. 3g-h.

Second Response: *This information is already contained in the paper. The legend and methods explain that both GFP+ and GFP- cells were FACS sorted based on GFP expression. They were obtained from the total luminal population purified as CD45-neg/CD31-neg/CD24-high/CD49f-low. Thus none of these subsets (i.e. GFP+ vs GFP-*

vs total) contain basal or stromal cells.

4) Blimp1 functions as a transcriptional repressor. However, the expression of downstream Blimp1 targets was not analyzed in this study. The authors need to examine some molecular role(s) of Blimp1 for mammary gland development and homeostasis.

Initial Response: *Here we exploit Blimp1 as a marker of an extremely rare stem cell population. Indeed the Blimp1-expressing cell population constitutes only ~0.2% of the total luminal population. Blimp-1 ChIP Seq analysis previously reported by ourselves and others exploited GFP-tagged knock-in alleles in combination with proven ChIP-grade anti-GFP mAb (Mould et al, 2015; Magnusdottir et al., 2013). Genome-wide high throughput methodologies would be technically impossible here.*

Examining whether Blimp1 overexpression suppresses Elf5 (or ER/PgR) expression in cultured cells would support the authors' notion of Blimp1 in this manuscript.

Second Response: *We never express the "notion" of any functional relationship between Blimp1 and Elf5 expression – indeed the Reviewer has failed to appreciate that the Blimp1-derived progeny all express Elf5, moreover no suitable Elf5+ERα- luminal cell population has ever been derived as a stable cell line, and finally we and many others have never succeeded in over-expressing Blimp1 in any cultured cell line (this observation has been published by Azim Surani's lab).*

Minor points 1) In Fig.1b, labelling with Krt8 and Krt14 is missing. 2) In manuscript page 8, line 1-2, the figure numbers are missing. 3) In Fig.2d, 2e, 3f, 4c, 4d and 4f, what types of cells were composed of the population is not mentioned.

Initial Response: *We thank the reviewer for pointing out these omissions. The errors have all been corrected, and missing Figure numbers inserted into the text.*

Please make sure that some numbers in the manuscript do not show in the figures: for instance, where do "~59% of total" (page 9, line 214) come from?

Second Response: *The number of 59% is arrived at by simple subtraction – it would be confusing to add this information to Figure 3, it belongs in the text.*